# The importance of crystalline phases in ice nucleation by volcanic ash

Elena C. Maters[1], Donald B. Dingwell[2], Corrado Cimarelli[2], Dirk Müller[2], Thomas F. Whale[1,3], Benjamin J. Murray[1]

[1]School of Earth and Environment, University of Leeds, Leeds LS2 9JT, United Kingdom
[2]Department of Earth and Environmental Sciences, Ludwig-Maximilians University, 80333 Munich, Germany
[3]School of Chemistry, University of Leeds, Leeds LS2 9JT, United Kingdom

*Correspondence to*: Elena C. Maters (e.c.maters@leeds.ac.uk)

**Abstract.** Volcanic ash is known to nucleate ice when immersed in supercooled water droplets. This process may impact the properties and dynamics of the eruption plume and cloud, as well as those of meteorological clouds once the ash is dispersed in the atmosphere. However, knowledge of what controls the ice-nucleating activity (INA) of ash remains limited, although it has been suggested that crystalline components in ash may play an important role. Here we adopted a novel approach using nine pairs of tephra and their remelted and quenched glass equivalents to investigate the influence of chemical composition, crystallinity and mineralogy on ash INA in the immersion mode. For all nine pairs studied, the crystal-bearing tephra nucleated ice at warmer temperatures than the corresponding crystal-free glass, indicating that crystalline phases are key to ash INA. Similar to findings for desert dust from arid and semi-arid regions, the presence of feldspar minerals characterises the four most ice-active tephra samples, although a high INA is observed even in the absence of alkali feldspar in samples bearing plagioclase feldspar and orthopyroxene. There is evidence of a potential indirect relationship between chemical composition and ash INA, whereby a magma of felsic to intermediate composition may generate ash containing ice-active feldspar or pyroxene minerals. This complex interplay between chemical composition, crystallinity, and mineralogy could help to explain the variability in volcanic ash INA reported in the literature. Overall, by demonstrating the importance of crystalline phases in the INA of ash, our study contributes insights essential for better appraising the role of airborne ash in ice formation. Among these is the inference that glass-dominated ash emitted by the largest explosive volcanic eruptions might be less effective at impacting ice-nucleating particle populations than crystalline ash generated by smaller, more frequent eruptions.

## 1 Introduction

Volcanic ash produced by explosive eruptions can act as ice-nucleating particles (INPs), promoting heterogeneous freezing of supercooled water in the vertical eruption plume, the laterally dispersed eruption cloud, and the wider atmosphere (Isono et al., 1959a; 1959b; Hobbs et al., 1971; Rose et al., 2003). Ice formation in these contexts is poorly understood yet may exert a profound influence on eruption plume/cloud dynamics and electrification (e.g., Herzog et al., 1998; Cimarelli et al., 2016), sequestration of gaseous species (e.g., Textor et al., 2003; Guo et al., 2004a), and ash aggregation and sedimentation (e.g., Guo et al., 2004b; Van Eaton et al., 2015). Ice formation on airborne ash can also affect atmospheric cloud properties and lifetime (e.g., Komabayasi, 1957; Seifert et al., 2011), and thereby the hydrological cycle and climate (e.g., Isono and Komabayasi, 1954). Ongoing volcanic activity generates a recurrent flux of ash particles into the atmosphere (176-256 Tg a$^{-1}$; Durant et al., 2010), whereas sporadic large eruptions can result in ash loadings greatly exceeding annual averages over very short (hour to day) time scales and transiently dominating INP populations (e.g., Isono et al., 1959a; 1959b; Hobbs et al., 1971).

By definition, volcanic ash consists of pyroclastic particles <2 mm in diameter, and is comprised of aluminosilicate glass as well as aluminosilicate and/or Fe(-Ti) oxide minerals (Heiken and Wohletz, 1992). The chemical composition of ash predominantly reflects the state of the source magma at the point of eruption but can also be influenced by lithic material (pre-existing country rock) entrained during the eruption (Heiken and Wohletz, 1992). As magma ascends to the surface, the aluminosilicate melt typically carries a cargo of mineral species in the form of crystals suspended within and originating from

the melt and/or from the surrounding country rock. Accordingly, upon magma fragmentation, the ash generated comprises a mixture of glass and crystal components (Heiken and Wohletz, 1992). The crystallinity refers to the relative abundance of crystals in ash (i.e., crystal mass/total mass) and typically ranges from 0 to 65 wt.%, depending on factors such as the prior state of the magma (e.g., chemical composition, temperature) and even the dynamics of the conduit (e.g., magma ascent rate; Heiken and Wohletz, 1992; Wright et al., 2012). The incorporation of lithic material from the conduit or vent and/or the interaction with ground or surface water can also influence the crystallinity of the bulk erupted ash. The mineralogy refers to the identities and abundances of crystalline phases in ash. Among the factors that influence crystallisation from the melt; the chemical composition of the magma is a key determinant of the mineral phases that can form (Heiken and Wohletz, 1992). Common mafic minerals in basaltic ash include pyroxene, olivine, amphibole, and (Ca-rich) plagioclase feldspar, whereas felsic minerals in rhyolitic ash include quartz, mica, amphibole, (Na-rich) plagioclase feldspar, and (K-rich) alkali feldspar (Fig. 1a; Rogers, 2015).

Field and laboratory measurements present conflicting evidence as to the ice-nucleating activity (INA) of ash (e.g., Isono et al., 1959a; Hobbs et al., 1971; Schnell and Delany, 1976; Schnell et al., 1982), even for samples from the same volcano, and it is far from clear what drives this variation (Durant et al., 2008). In immersion freezing experiments, Soufrière Hills ash has been found to range from inactive to highly active in nucleating ice, with the discrepancy inferred to relate to differences in ash composition and sample preparation methods (Schill et al., 2015; Mangan et al., 2017; Jahn et al., 2019). Studies on desert dust from arid and semi-arid regions (1000-3000 Tg $a^{-1}$ emitted; Penner et al., 2001) - considered one of the most important INP types globally (Hoose et al., 2010; Vergara-Temprado et al., 2017) - suggest that chemical composition, crystallinity and mineralogy can influence the abundance of ice-active surface sites on the solid particles (Murray et al., 2012, and references therein). Specifically, the presence of K-rich feldspar is thought to dominate the INA of dust (Atkinson et al., 2013; Yakobi-Hancock et al., 2013; Kaufmann et al., 2016). There is increasing evidence that similar factors may influence ice nucleation by volcanic ash. Kulkarni et al. (2015) argued that the presence of amorphous material reduced the INA of Eyjafjallajökull ash compared to Arizona test dust, based on the notion that crystalline structures provide preferred configurations for water molecules to bind at the particle surface (Pruppacher and Klett, 2010). Schill et al. (2015) proposed that, aside from amorphous versus crystalline content, differences in mineralogy could explain the INA of ash from Soufrière Hills, Fuego, and Taupo volcanoes. Recently, Jahn et al. (2019) suggested that feldspar and pyroxene minerals were responsible for the high INA of Soufrière Hills, Fuego, and Santiaguito ash samples. Genareau et al. (2018) conversely noted a broad trend between chemical composition and ice nucleation, with the INA of five ash samples increasing with $K_2O$ content and decreasing with MnO content. To date however, the roles and potential interplay of differing physicochemical attributes in determining a solid particle's INA remain poorly understood, having rarely been systematically investigated for any ice-nucleating material let alone volcanic ash.

Here we examine the influence of three properties dictated primarily by the state of the erupted source magma - chemical composition, crystallinity, and mineralogy - on the INA of volcanic ash in the immersion mode, which is likely relevant to ash particles in the water-rich eruption plume/cloud and in mixed-phase atmospheric clouds (Textor et al., 2006; McNutt and Williams, 2010; Pruppacher and Klett, 2010; Murray et al., 2012). To assist in disentangling the individual effects of these properties on ice nucleation, we have adopted a novel approach of sample selection by using a wide range of natural tephra *and* their remelted and quenched glass equivalents. By finding that crystalline phases promote ash ice nucleation and that magma composition may exert an indirect effect via its influence on ash mineralogy, we contribute to an improved understanding of the potential for airborne ash from different eruptions to impact ice formation above the volcanic vent and/or once dispersed in the ambient atmosphere.

## 2 Materials and methods

### 2.1 Volcanic tephra and glass samples

Eighteen tephra and (remelted and quenched) glass powders spanning a range of compositions associated with volcanic activity were studied (Table 1). The powders are represented by an abbreviated sample code with the subscripts 'teph' or 'glass' used to designate tephra or glass material, respectively. The tephra samples correspond to ash or pumice originating from different eruptions. The glass samples were synthesised by melting a portion of the tephra at 1400 to 1600 °C, homogenising the melts by stirring for 12 to 72 h, and quenching the melts at room temperature to form glasses. This technique for generating volcanic glass has been used previously in studies of volcanic ash reactivity (e.g., Ayris et al., 2013; 2014; Maters et al., 2016; 2017). All samples were crushed to fine powders in a ball mill using a zirconia ceramic ball and vial to ensure consistent treatment of the tephra and glass materials prior to ice nucleation experiments. This also reduced the influence of chemically-altered surfaces resulting from tephra interaction with gases and/or liquids post-eruption (e.g., Delmelle et al., 2007), by exposing fresh surfaces with chemical and mineralogical properties reflective of the source magma and any entrained lithic material. This allowed us to address the study objective of assessing specifically the role of chemical composition, crystallinity, and mineralogy in ice nucleation by volcanic ash. Aside from crushing, the samples were not processed by rinsing with water or otherwise, to avoid further alteration of these materials on short time scales (e.g., exposure to water is known to change the INA of some minerals; Harrison et al., 2016; Kumar et al., 2018). The specific surface area ($SSA_{BET}$; Table 1) of the samples after overnight degassing was obtained from a ten-point $N_2$ adsorption isotherm at -196 °C based on the Brunauer, Emmet and Teller model (Brunauer et al., 1938) using a Micromeritics TriStar 3000 instrument.

The chemical composition of the tephra and glass samples (Table S1) was measured by X-ray fluorescence at 3.2 kW ionisation energy using a Philips Analytical MagiX PRO spectrometer. The conventional classification of these materials according to total alkali ($Na_2O + K_2O$) versus silica ($SiO_2$) content is shown on a Total Alkali versus Silica diagram in Fig. 1b (Le Maitre et al., 2002). The proportions of various crystalline phases in the tephra samples (Table 2) were determined by X-ray diffraction (XRD) with a $Cu_{K\alpha1}$ X-ray beam using a GE diffractometer (XRD 3003 TT) and the Profex software program for Rietveld refinement (Döbelin and Kleeberg, 2015). Briefly, this involved crushing the tephra and spiking it with a known mass (~17 wt.%) of pure Si powder, as crystalline internal standard, to quantify the crystallinity and mineralogy of the samples. Note that in this study, we use the terms 'alkali' and 'plagioclase' to refer specifically to K-rich and Na-/Ca-rich feldspars, respectively. While the $LIP_{teph}$ and $CID_{teph}$ crystallinities cannot be quantified below the ~2 wt.% limit of the technique, this does not rule out the possibility of smaller amounts of crystals and/or nanoscale crystallites being present in these samples. These minor components are listed along with the quantitative mineralogy of the tephra samples in Table 2. The glasses were also analysed by XRD to confirm their amorphous nature (i.e., the absence of crystalline minerals within the ~2 wt.% quantification limit).

### 2.2 Immersion mode ice nucleation experiments

The INA of the tephra and glass samples was assessed using a microlitre Nucleation by Immersed Particles Instrument (µL-NIPI). This instrument has been described in detail elsewhere (Whale et al., 2015) and has been used previously to study heterogeneous ice nucleation by mineral and ash material (e.g., Atkinson et al., 2013; Harrison et al., 2016; Mangan et al., 2017; Whale et al., 2017). Briefly, a 1 wt.% sample suspension in Milli-Q water (18.2 MΩ·cm) was shaken for a few minutes by a vortex mixer, and then pipetted in an array of 30 to 40 1 µL droplets onto a hydrophobic silanised glass cover slip placed on a temperature-controlled stage (Grant-Asymptote EF600 Stirling Cryocooler). The stage was cooled from room temperature at a rate of -5 °C min$^{-1}$ down to 0 °C, and subsequently at a rate of -1 °C min$^{-1}$ until all droplets were frozen. A dry nitrogen flow (~0.2 L min$^{-1}$) over the droplets prevented condensation and frost accumulation on the cover slip and hence served to

avoid frozen droplets affecting neighbouring liquid droplets. A digital camera was used to observe the droplets throughout the experiment and determine the fraction of droplets frozen as a function of temperature $f_{ice}(T)$ according to:

$$f_{ice}(T) = n_{ice}(T)/n, \qquad (1)$$

where $n_{ice}(T)$ is the cumulative number of droplets frozen at temperature ($T$) and $n$ is the total number of droplets in the experiment. At least three replicate experiments were conducted for each sample. In addition, ice nucleation of the background water at higher temperatures than those predicted by classical nucleation theory (Murray et al., 2010; Koop and Murray, 2016), due to impurities in the water and/or effects of the cover slip (Polen et al., 2018), was assessed by acquiring baseline droplet freezing measurements of water containing no added particles.

To facilitate comparison of different materials including across literature studies, their ability to nucleate ice is often expressed in terms of the ice nucleation active site density $n_s(T)$, which represents the number of active sites per unit surface area of a solid sample on cooling from 0 °C down to temperature ($T$) (Connolly et al., 2009):

$$\frac{n_{ice}(T)}{n} = 1 - \exp(-n_s(T)A), \qquad (2)$$

where $A$ is the total surface area of the solid sample per droplet. Although the fundamental nature of ice-active sites remains unclear, and may vary across different materials, $n_s(T)$ allows us to empirically define the INA of a range of solid substrates (Vali, 2014). The uncertainty in $n_s(T)$ was calculated using simulations of possible active site distributions propagated with the uncertainty in surface area of nucleant per droplet, as outlined in Harrison et al., (2016). The uncertainty in temperature for the µL-NIPI is estimated to be ±0.4 °C (Whale et al., 2015).

## 3 Results

The $f_{ice}(T)$ and $n_s(T)$ values for the eighteen samples studied are shown in Fig. 2. For the sake of clarity, droplet freezing events across replicate experiments have been combined here to generate a single dataset for each sample. The samples display wide variation in freezing temperatures with generally higher values associated with the tephra (-3 to -25 °C; Fig. 2a, c) than with the glass (-12 to -30 °C; Fig. 2b, d).

For the tephra samples, the $f_{ice}(T)$ curves are separate from those of the background water (Fig. 2a). This gives confidence to attribution of the observed freezing to ice nucleation by tephra particles. In terms of ice nucleation active site densities (Fig. 2c), taking the temperature at which $n_s \approx 1$ cm$^{-2}$ as a simple single-number proxy for INA ($T_{n_s \approx 1\,cm^{-2}}$), the most active tephra are the trachyphonolite samples NUO$_{teph}$ and AST$_{teph}$ with $T_{n_s \approx 1\,cm^{-2}}$ values of -5.0 °C and -6.4 °C, respectively, followed by the andesite samples COL$_{teph}$ and TUN$_{teph}$ with $T_{n_s \approx 1\,cm^{-2}}$ values of -8.1 °C and -10.5 °C, respectively. The least active tephra are the basalt and phonolite samples KIL$_{teph}$ and LAC$_{teph}$ with $T_{n_s \approx 1\,cm^{-2}}$ values of -17.5 °C and -17.9 °C, respectively.

For the glass samples in contrast, there is significant overlap of temperatures at which their $f_{ice}(T)$ curves and those of the background water fall, spanning a range from -18 °C to -35 °C (Fig. 2b). As illustrated by the overlap of error bars in their $n_s(T)$ curves (Fig. 2d), this prevents attribution of the observed freezing to ice nucleation by glass particles and comparison of individual glass activities. It also likely explains the poorer reproducibility seen in replicate experiments of the glass material relative to the tephra material (Fig. 3). Hence, in most cases the reported $n_s(T)$ values should be regarded as upper limits. However, the trachyte and andesite samples CID$_{glass}$ and COL$_{glass}$ stand out with only partial overlap of $f_{ice}(T)$ curves and the freezing temperature range of the background water (Fig. 2b), and thus are identified as being the most active glasses in terms of ice nucleation active site densities (Fig. 2d), with $T_{n_s \approx 1\,cm^{-2}}$ values of -16.8 °C and -17.0 °C, respectively.

Consideration of the compositionally analogous tephra-glass pairs clearly illustrates the observation of tephra nucleating ice more effectively than the equivalent remelted and quenched glass (Fig. 3). The $n_s(T)$ curves of each tephra sample fall at higher temperatures compared to those of its counterpart glass sample, albeit displaying varying temperature differences between them.

## 4 Discussion

The eighteen samples were chosen to encompass a variety of chemical compositions, crystallinities and mineralogies encountered in volcanic ash, with the aim of investigating the influence of these physicochemical properties on ash INA. The results of our ice nucleation experiments are examined in relation to each of these properties below.

### 4.1 Crystallinity

As noted above, our sample pairs of crystal-bearing tephra and crystal-free glass of nearly identical chemical composition (Table S1) constitute a unique approach to studying controls on volcanic ash ice nucleation. These pairs were chosen to disentangle variation in crystallinity from that in composition, which together might complicate interpretation of INA trends in natural ash collections. For all pairs studied, the tephra nucleates ice at higher temperatures than the corresponding glass (Fig. 3), overall implying a positive effect of crystals on INA. Even the dominantly glassy LIP$_{teph}$ and CID$_{teph}$ (<2 wt.% crystallinity) display higher INA than their counterpart LIP$_{glass}$ and CID$_{glass}$, which could reflect the influence of minor crystalline components (below quantification by XRD; Table 2) in these tephra on their ability to nucleate ice. However, the difference between tephra and glass $T_{n_s \approx 1\,cm^{-2}}$ values across the nine sample pairs ($\Delta T_{n_s \approx 1\,cm^{-2}}$, ranging from ~2 to 19 °C) does not vary simply with respect to tephra crystallinity (<2 to 66 wt.%; Fig. S1a). Additionally, a plot of $T_{n_s \approx 1\,cm^{-2}}$ values versus crystallinity of the tephra samples shows no correlation between ice nucleation and crystalline content (Fig. S1b). The compositionally quite similar NUO$_{teph}$ and AST$_{teph}$ display the highest INA ($T_{n_s \approx 1\,cm^{-2}}$ values of -5.0 and -6.4 °C, respectively) and are characterised by markedly contrasting crystallinities (60 and 28 wt.%, respectively). A difference in crystallinity was proposed as one of a number of potential explanations for the variable INA of two ash samples from Soufrière Hills volcano, with the 100% crystalline material produced by a dome collapse showing a higher INA than the 89% crystalline material produced by a magmatic eruption (Schill et al., 2015; Mangan et al., 2017). However, in light of our findings, it seems unlikely that this slight difference in crystallinity of two Soufrière Hills ash samples can adequately explain the large disparity in their INA.

A study comparing ice nucleation by crystalline and glassy anorthite ($CaAl_2Si_2O_8$), with the former displaying a $n_s(T)$ curve reaching higher freezing temperatures than the latter, suggested that crystals may introduce rarer, more ice-active surface sites but are not required for ice nucleation (Harrison et al., 2016). In contrast, our observations strongly suggest that the presence of crystals is crucial in making volcanic ash an effective ice nucleant, although the abundance of crystalline phases in ash may be less important than the mere presence and specific properties of those phases in determining the INA.

### 4.2 Mineralogy

Consideration of the nine tephra samples may provide insight into the influence of mineralogy on the INA of volcanic ash. As noted above, comparison of the compositionally analogous tephra-glass pairs points to a role of crystalline phases in promoting freezing, and we infer that the properties of those phases have a strong effect on a sample's INA. A study of ash from Soufrière Hills, Fuego, and Taupo volcanoes attributed differences in their INA to their contrasting mineralogies (Schill et al., 2015). More recently, Jahn et al. (2019) proposed that feldspar and pyroxene were responsible for ice nucleation by ash from Soufrière Hills, Fuego, and Santiaguito volcanoes. Overall, when the $T_{n_s \approx 1\,cm^{-2}}$ values are plotted against the content of various minerals

in the tephra samples studied here, no clear correlations are evident (Fig. 4). However, certain features do stand out; the two most ice-active $NUO_{teph}$ and $AST_{teph}$ have the highest contents of alkali feldspar (60 and 19 wt.%, respectively; Fig. 4a), while the next most ice-active $COL_{teph}$ and $TUN_{teph}$ are characterised by an abundance of plagioclase feldspar (55 and 43 wt.%, respectively; Fig. 4b) and lesser amounts of orthopyroxene (7 and 5 wt.%, respectively, Fig. 4d).

The $n_s(T)$ curves of our tephra samples are compared with the INA of alkali (K-) and plagioclase (separated into Na-/Ca- and Na-) feldspars in Fig. 5. The $n_s(T)$ curves of $NUO_{teph}$ and $AST_{teph}$ span temperatures consistent with the K-feldspar parameterisation compiled from literature data (Harrison et al., in prep.), supporting the notion that the INA of these two samples relates to the presence of this mineral phase. The next two most ice-active materials $COL_{teph}$ and $TUN_{teph}$ contain no appreciable quantity of alkali feldspar, instead being rich in plagioclase feldspar. However, the $n_s(T)$ curves of these two

samples are inconsistent with the relatively low INA of Na-/Ca-feldspar reported in the literature (Fig. 5). This may point to the presence in $COL_{teph}$ and $TUN_{teph}$ of ice-active plagioclase feldspar characterised by an INA closer to the Na-feldspar (albite) parameterisation, or potentially more akin to the hyper-active feldspars measured by Harrison et al. (2016; Fig. 5). It is not clear why these hyper-active samples (Amelia albite and TUD#3 microcline) have a much greater INA relative to the majority of feldspars tested (Harrison et al., 2016; Peckhaus et al., 2016), but such wide variability may relate to the specific mechanisms

and/or conditions of formation and subsequent processing of individual samples (Welti et al., 2019), and it might be that plagioclase feldspar in $COL_{teph}$ and $TUN_{teph}$ was produced in a way that gives rise to enhanced activity. Alternatively, the high INA of these two tephra samples may relate to the influence of some other mineral component such as orthopyroxene; this possibility is discussed in more detail below.

    Other tephra samples containing feldspar are comparatively less effective at nucleating ice, in particular; the intermediately

ice-active $ETN_{teph}$ (44 wt.% plagioclase feldspar; Fig. 4b) and the least ice-active $KIL_{teph}$ (3 wt.% plagioclase feldspar; Fig. 4b) and $LAC_{teph}$ (9 wt.% alkali feldspar; Fig. 4a). Such differences might relate to the specific chemistry of the mineral phases present in the tephra (Zolles et al., 2015; Welti et al., 2019). Harrison et al. (2016) showed that the INA of feldspar generally decreases from the K end-member ($KAlSi_3O_8$) to the Na end-member ($NaAlSi_3O_8$) to the Ca end-member ($CaAl_2Si_2O_8$), with the exception of a hyper-active Amelia albite specimen (Fig. 5). Based on electron microprobe analysis (Text S1, Table S2),

the $Na_2O/CaO$ ratio in plagioclase feldspar in $COL_{teph}$ and $TUN_{teph}$ (both ~0.5) is higher than in $ETN_{teph}$ and $KIL_{teph}$ (both ~0.2), reflecting a greater proportion of the more ice-active $NaAlSi_3O_8$ relative to $CaAl_2Si_2O_8$ in the former samples. On the other hand, the $K_2O/Na_2O$ ratio in alkali feldspar in $NUO_{teph}$, $AST_{teph}$ and $LAC_{teph}$ (1.2, 5.0, 1.9, respectively) does not support a link between these samples' INA and the proportion of $KAlSi_3O_8$ relative to $NaAlSi_3O_8$. This is consistent with the results of Whale et al. (2017), who found that the INA of alkali feldspar does not relate directly to K content, but rather to the presence

of perthitic intergrowth microtexture arising from phase separation (exsolution) into Na- and K-rich regions. Strain at the boundary of these regions gives rise to nanoscale topographic features that are suggested to be important in generating sites for ice nucleation (Whale et al., 2017; Holden et al., 2019). It may be that these features stabilise patches of the high-energy (100) crystallographic plane exposed by surface defects, which Kiselev et al. (2017) showed to be favourable sites for ice nucleation on alkali feldspar.

However, perthite in alkali feldspar develops in metamorphic and plutonic contexts during slow cooling at temperatures <700 °C (Parsons, 2010), and is generally not expected in volcanic ash which cools rapidly from magmatic down to ambient temperatures during eruption (Parsons et al., 2015). An absence of perthitic microtexture is consistent with the low INA of $LAC_{teph}$ in spite of its alkali feldspar content (9 wt.%). This is supported by evidence that the alkali feldspar mineral sanidine sourced from the same geological setting as $LAC_{teph}$ (Eifel volcanic field) lacks perthitic texture and exhibits a poor ability to

nucleate ice (Whale et al., 2017). A recent study similarly found volcanic sanidine from Germany to be the least ice-active among the alkali feldspar samples tested (Welti et al., 2019). In contrast, an absence of perthitic microtexture is inconsistent with the high INA of $NUO_{teph}$ and $AST_{teph}$, and perhaps some other textural feature underlies these samples' ability to nucleate

ice as effectively as alkali feldspar of non-pyroclastic origin (Fig. 5). Pyroclastic material from both the 1538 Monte Nuovo eruption (i.e., the origin of $NUO_{teph}$) and the ~4 ka Astroni eruption (i.e., the origin of $AST_{teph}$) has been found to contain anti-rapakivi overgrowth microtexture characterised by plagioclase feldspar cores rimmed by alkali feldspar (D'Oriano et al., 2005; Astbury et al., 2016; 2018). We are not aware of any studies reporting similar textures in pyroclastic material from the 12.9 ka Laacher See eruption (i.e., the origin of $LAC_{teph}$). Such textures are challenging to resolve optically in powdered samples including the tephra studied here. Further optical and microanalytical (Scanning- and Transmission Electron Microscopy) observations (e.g., Whale et al., 2017; Holden et al., 2019) will be needed to explore whether the boundary between Na- and K-rich regions in anti-rapakivi microtexture may give rise to nanoscale topography that induces effects analogous to perthitic microtexture in promoting ice nucleation.

In addition to feldspar, several of the tephra samples contain pyroxene (Table 2), an aluminosilicate mineral group of the general formula $XYZ_2O_6$, where X and Y are often $Mg^{2+}$, $Fe^{2+}$ or $Ca^{2+}$ and Z is $Si^{4+}$ or sometimes $Al^{3+}$ (Morimoto et al., 1988). A solid solution exists between the $Mg_2Si_2O_6$ and $Fe_2Si_2O_6$ end-members with small amounts of $Ca^{2+}$ substitution possible (orthopyroxenes), whereas solid immiscibility occurs between other compositions particularly with higher $Ca^{2+}$ content (clinopyroxenes). The presence of orthopyroxene distinguishes $COL_{teph}$ and $TUN_{teph}$ from the other tephra (Fig. 4d), raising the question of whether it may underlie the high INA of these two samples (i.e., rather than plagioclase feldspar). We are aware of a few early studies on ice nucleation by ortho- and clinopyroxene minerals (hypersthene, augite; Hama and Itoo, 1956; Isono and Ikebe, 1960), but these studies only report semi-quantitative onset freezing temperatures (between -8 and -15 °C). More recently, Jahn et al. (2019) measured the INA of a clinopyroxene specimen (freezing from -8 to -24 °C), citing its behaviour to explain the INA of three pyroxene-containing volcanic ash samples. However, XRD analysis indicated that this specimen comprising diopside-augite also contained ~5 wt.% feldspar, which might have influenced the INA observed. In any case, it should be emphasised that a single mineral specimen might not provide a good representation of the INA of a given mineral type, as shown by studies on ice nucleation by feldspar and quartz (Harrison et al., 2016; Whale et al., 2017; Harrison et al., in prep.). Therefore, at present we cannot rule out a potential influence of pyroxene on ice nucleation by volcanic ash. Additional research is needed to quantify the INA of a range of pyroxene minerals, and probe the nature of their ice-nucleating properties, in order to better inform this assessment.

### 4.3 Chemical composition

To explore a potential link between chemical composition and INA, the $T_{n_s \approx 1\,cm^{-2}}$ values of the tephra and glass samples are plotted as a function of $SiO_2$, $Al_2O_3$, $Fe_2O_3$, MgO, CaO, $Na_2O$, $K_2O$, $TiO_2$, MnO and $P_2O_5$ contents in Fig. S2. No clear correlations are observed in any of these scatter plots to indicate a compositional dependency of the tephra or glass INA. This stands in apparent contrast to the recent work of Genareau et al. (2018) (based on a sample set of two rhyolites and three basalts), who reported that the $n_s(T)$ of volcanic ash correlates positively with $K_2O$ content at -25 °C and negatively with $TiO_2$ and MnO contents from -30 to -35 °C.

The glass samples, in lacking crystalline minerals, are well-suited to assess for any direct relationships between INA and specific element oxide abundances. However, due to the overlap of droplet freezing temperatures of the glass suspensions and the background water (Fig. 2b), our ability to distinguish differences in INA across the nine samples is impeded. While $CID_{glass}$ and $COL_{glass}$ are the most ice-active glass samples, with signals clearly above the background (Fig. 2b, d), they represent intermediate chemical compositions and thus their behaviour does not support any simple link between ash INA and chemical composition. It is possible that $CID_{glass}$ and $COL_{glass}$ contain very small amounts of crystals below detection by XRD that survived melting or formed during quenching, which could explain why these glasses stand out in their ability to nucleate ice.

In contrast, the tephra samples are characterised by variations in crystallinity and mineralogy as well as composition, which convolutes the assessment of relationships between INA and specific element oxide abundances. However, if the crystallinity

and mineralogy of the tephra samples are taken into consideration, a broad pattern emerges in the plots of $T_{n_s \approx 1\ cm^{-2}}$ versus $Fe_2O_3$, MgO, and CaO contents (Fig. S2c-e). Excluding a cluster of three samples with comparatively low crystallinities (LIP$_{teph}$, CID$_{teph}$, LAC$_{teph}$), the $T_{n_s \approx 1\ cm^{-2}}$ decreases with increasing $Fe_2O_3$, MgO, and CaO contents for NUO$_{teph}$, AST$_{teph}$, COL$_{teph}$, TUN$_{teph}$, ETN$_{teph}$, and KIL$_{teph}$ (Fig. 6), in an order consistent with interpretations relating to their feldspar contents and chemistries and/or a potential effect of orthopyroxene in two of these samples (see discussion Sect. 4.2). This conforms to the notion of an indirect relationship between chemical composition and volcanic ash INA, whereby $FeO/Fe_2O_3$, MgO, and CaO contents increase from felsic to mafic magma, influencing the mineral phases that can crystallise from the magma and hence exist in the resultant ash (Fig. 1b).

## 5 Conclusions and implications

Here we used nine compositionally analogous pairs of natural tephra and remelted and quenched glass to investigate the influence of chemical composition, crystallinity and mineralogy on the INA of volcanic ash. The higher INA of the tephra relative to the glass strongly suggests that the presence of crystalline phases promotes ice nucleation. The large variability in INA of the tephra is inferred to reflect an influence of mineralogy - and hence an indirect influence of magma composition - on ice nucleation. As in desert dust, alkali feldspar is probably the most ice-active component in volcanic ash, conferring the highest INA to NUO$_{teph}$ and AST$_{teph}$ in this study. However, the ability of alkali feldspar in ash to nucleate ice likely cannot unequivocally be attributed to perthitic microtexture, as has been done for alkali feldspar of non-pyroclastic origin. Additional research is needed to explore whether other textural features in ash may elicit a similar effect in promoting ice nucleation. Further, the presence of alkali feldspar is neither always sufficient nor necessary for effective ice nucleation by ash. The high INA of COL$_{teph}$ and TUN$_{teph}$ may alternatively reflect very ice-active plagioclase feldspar, or possibly orthopyroxene, which is present exclusively in these studied tephras. Previous studies on Soufrière Hills ash, also lacking alkali feldspar and containing plagioclase feldspar and orthopyroxene, have reported low to high INA of this ash (Schill et al., 2015; Mangan et al., 2017; Jahn et al., 2019). Future studies quantifying the INA of individual crystalline phases found in ash will be necessary to unravel the precise role of mineralogy in volcanic ash ice nucleation.

An improved knowledge of the link between particular ash properties and ash INA may ultimately enhance predictive capability regarding volcanic eruptions likely to generate ice-active material. For example, as crystalline phases are primarily controlled by magma composition and storage/ascent conditions (Rogers, 2015), we speculate that highly ice-active ash particles might be erupted by volcanoes with intermediate to felsic alkaline magmas giving rise to feldspar crystals featuring overgrowth textures (e.g., Astbury et al., 2016; 2018) or potentially pyroxene crystals with high INA for reasons yet unknown (e.g., Jahn et al., 2019). In addition, an eruption producing an abundance of crystal-bearing particles is expected to elicit a greater impact on heterogeneous ice nucleation than an eruption producing an abundance of crystal-free glass particles, all else being equal. Accordingly, massive outputs from the largest and most explosive eruptions, corresponding to violent caldera-forming ignimbrite events that generate ash clouds dominated by the glassy component (Sparks et al., 1997; Cather et al., 2009), might be less efficient in affecting INP populations than ash emissions from smaller eruptions. Further, since airborne ash typically becomes enriched in glassy fragments during long-range transport due to earlier gravitational settling of crystalline fragments (Hinkley et al., 1982), the INA of a suspended ash population is expected to decrease over time and distance from the volcano.

Lastly, it must be noted that once ash particles are generated, their surface properties can be altered by interactions with gases and condensates (e.g., $H_2O$, $SO_2$, $H_2SO_4$, HCl, HF) at variable temperatures in the eruption plume/cloud and ambient atmosphere (Delmelle et al., 2007; Ayris et al., 2013; 2014; Maters et al., 2016; 2017). The effects of such interactions on ash INA are not known, although it has been suggested from field measurements that volcanic gases may deactivate INPs (Schnell

and Delany, 1976; Schnell et al., 1982). Laboratory studies on desert dust show that 'aging' of dust particle surfaces by exposure to $H_2SO_4$ vapours at elevated temperatures reduces dust INA, possibly by destroying ice-active surface sites (Sullivan et al., 2010; Niedermeier et al., 2011). Moreover, it has recently been shown that even very low concentrations of soluble salts ($\sim10^{-4}$ M) can influence the INA of feldspar minerals (Kumar et al., 2018; Whale et al., 2018), and we cannot exclude the possibility that small amounts of NaCl or KCl formed by prior ash-gas/condensate interactions in our tephra samples reduced their INA. However, given the strong correlations observed between INA and composition of the crystalline tephra samples (Fig. 6), we do not think that a potential influence of soluble salts on freezing temperatures affects the general conclusions of this study. Exploring such potential eruptive and atmospheric controls on ash INA is an important next step towards developing a better understanding of the capacity of volcanic ash emissions to affect heterogeneous ice nucleation during their airborne lifetime.

*Data availability*: e.c.maters@leeds.ac.uk

*Author contribution*: E.M. designed the study and carried out the experiments. D.D. and C.C. provided the tephra samples and produced the glass samples. D.M. performed chemical and mineralogical analyses of these samples. T.W. performed the Poisson Monte Carlo error calculations. B.M supervised the project and provided insight on data interpretation. E.M. wrote the manuscript with contributions from all co-authors.

*Competing interests*: The authors declare that they have no conflict of interest.

*Acknowledgements*: E.M. is funded by the European Union's Horizon 2020 Research and Innovation Programme under the Marie Skłodowska-Curie Actions grant agreement No. 746695 (INoVA project). B.M. also acknowledges the European Research Council (MarineIce: No. 648661; CryoProtect: No. 713664) for funding. The authors wish to thank Ulrich Küppers for collecting many of the tephra samples used in this study, and Bruce Houghton who made possible the collection of Kilauea achneliths. We also thank Nora Groschopf (Institute of Geosciences, Johannes Gutenberg University Mainz) for XRF analyses of our samples. We are also grateful to William Orsi for enabling access to ultrapure water facilities at LMU, Alex Harrison for providing the plagioclase and alkali feldspar parameterisations, and to Sebastien Sikora, Lesley Neve, Fiona Keay and Andy Connelly for laboratory assistance.

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

**Table 1.** Details of the volcanic tephra and glass samples used in this study.

| Sample code | Source volcano | Eruption date[a] | Classification[b] | SSA$_{BET}$[c] (tephra/glass) ($m^2\ g^{-1}$) |
|---|---|---|---|---|
| LIP$_{teph/glass}$ | Lipari (Italy) | 1230 | rhyolite | 1.8/1.1 |
| COL$_{teph/glass}$ | Colima (Mexico) | Jan-Feb 2017 | andesite | 1.9/0.9 |
| TUN$_{teph/glass}$ | Tungurahua (Ecuador) | Feb 2014 | andesite | 1.4/1.1 |
| CID$_{teph/glass}$ | Sete Cidades (Portugal) | 16 ka | trachyte | 1.6/1.4 |
| AST$_{teph/glass}$ | Astroni (Italy) | 3.8-4.4 ka | trachyphonolite | 3.7/1.4 |
| NUO$_{teph/glass}$ | Monte Nuovo (Italy) | Sept-Oct 1538 | trachyphonolite | 4.6/1.3 |
| LAC$_{teph/glass}$ | Laacher See (Germany) | 12.9 ka | phonolite | 3.3/0.9 |
| ETN$_{teph/glass}$ | Mount Etna (Italy) | July 2014 | trachybasalt | 1.7/1.1 |
| KIL$_{teph/glass}$ | Kilauea (Hawaii) | July 2018 | basalt | 2.1/1.1 |

[a]Refers to the eruption of origin of the tephra material. This does not apply to the glass material as it has been synthesised (from tephra) in the laboratory by a melting, homogenising and quenching protocol. [b]According to the Total Alkali versus Silica igneous rock classification diagram (Fig. 1) based on chemical composition (Table S1). [c]Uncertainty is in the range of 0.5-1.2 %.

**Table 2.** Crystallinity and mineralogy of the tephra samples used in this study, in wt.%.

| Sample[a] | Crystallinity | Alkali (K-rich) feldspar | Plagioclase (Na-/Ca-rich) feldspar | Clino-pyroxene | Ortho-pyroxene | Quartz | Fe(-Ti) oxide | Olivine |
|---|---|---|---|---|---|---|---|---|
| LIP$_{teph}$ | <2 | - | - | - | - | - | m.c. | - |
| COL$_{teph}$ | 62 | m.c. | 55 | - | 7 | m.c. | m.c. | - |
| TUN$_{teph}$ | 54 | - | 43 | 6 | 5 | - | m.c. | - |
| CID$_{teph}$ | <2 | m.c. | - | m.c. | - | - | m.c. | - |
| AST$_{teph}$ | 28 | 19 | 7 | 2 | - | - | m.c. | - |
| NUO$_{teph}$ | 60 | 60 | - | - | - | - | m.c. | - |
| LAC$_{teph}$ | 11 | 9 | - | - | - | 2 | - | - |
| ETN$_{teph}$ | 66 | - | 44 | 22 | - | - | m.c. | - |
| KIL$_{teph}$ | 3 | - | 3 | - | - | - | - | m.c. |

[a]Sample codes are listed in Table 1. m.c. = minor component; below ~2 wt.% quantification limit by XRD.

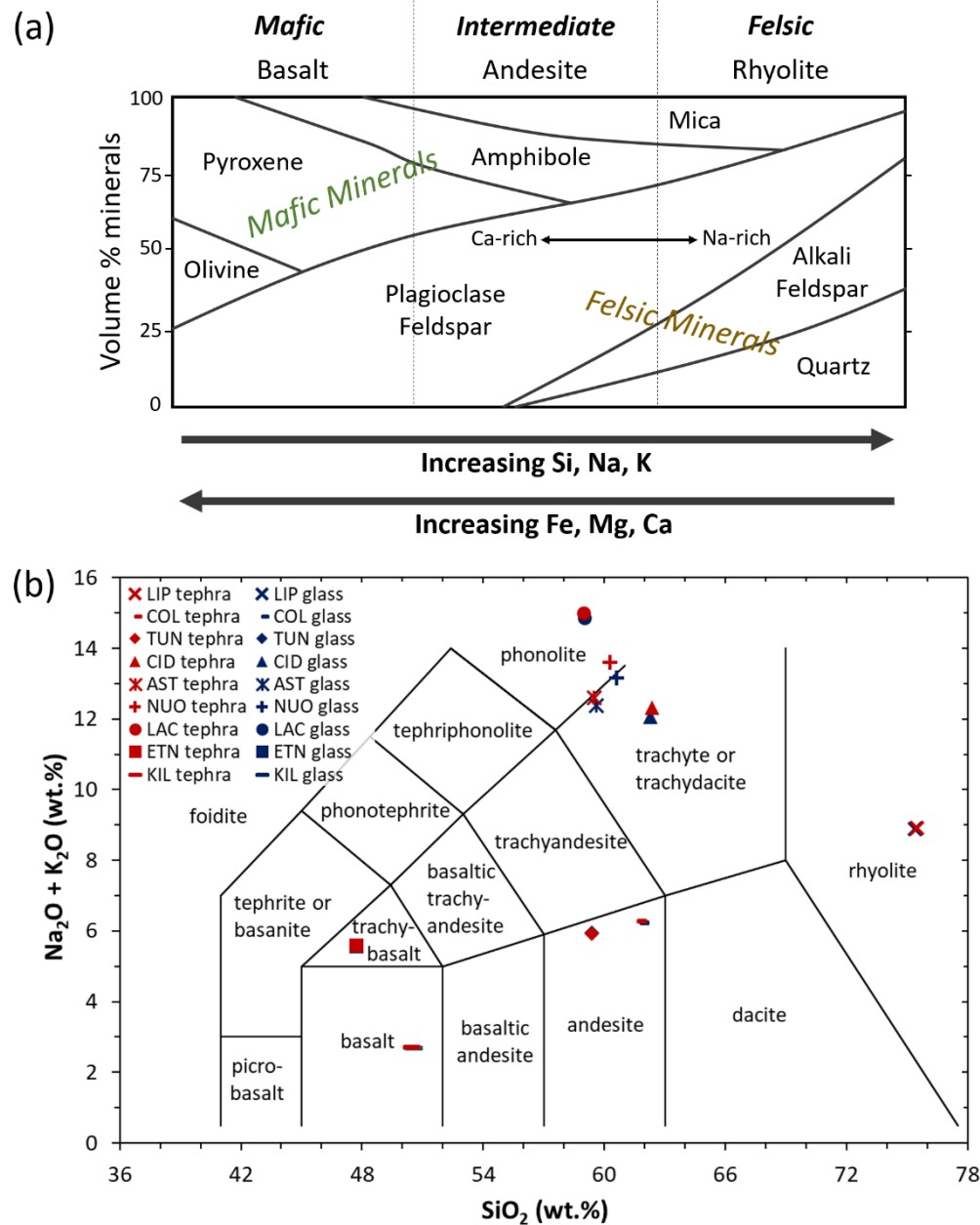

**Figure 1.** (a) Schematic summarising the mineralogy of common igneous rock types. Modified after Rogers (2015) (b) Total Alkali versus Silica diagram showing the classification of the tephra (red symbols) and glass (blue symbols) used in this study. Sample codes are listed in Table 1. Modified after Le Maitre et al. (2002).

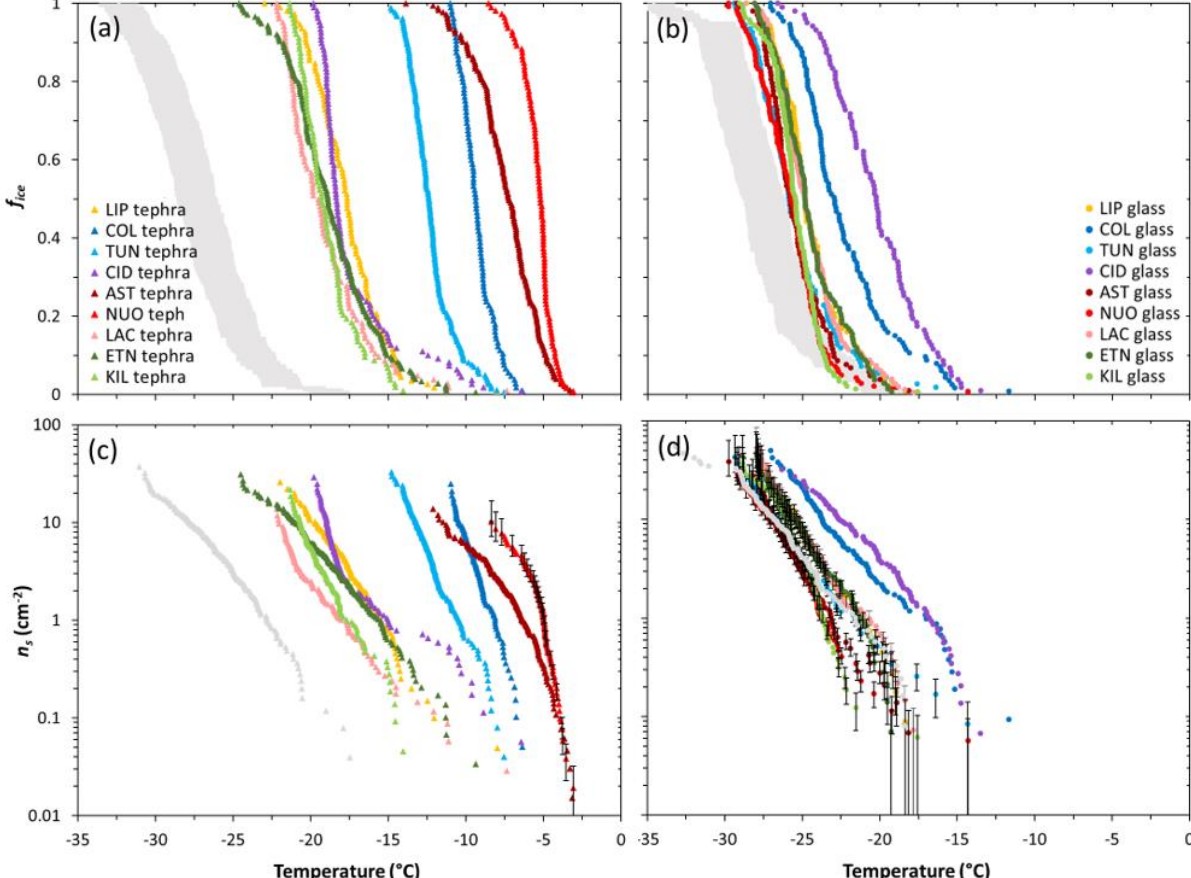

**Figure 2.** Droplet fraction frozen ($f_{ice}$) as a function of temperature for 1 wt.% suspensions of (a) tephra or (b) glass in water. The grey bands represent the spread of $f_{ice}(T)$ measurements (mean values ± standard deviation) of the background water (i.e., containing no added sample). Ice nucleation active site density ($n_s$) as a function of temperature for 1 wt.% suspensions of (c) tephra or (d) glass in water. For the sake of comparison, the grey curves represent theoretical upper limit $n_s(T)$ values of the background water, calculated using the upper limit $f_{ice}(T)$ measurements (mean values + standard deviation) of the background water, and assuming it contains particles with $SSA_{BET}$ values equal to the lowest from the tephra and glass sets (1.4 and 0.9 m$^2$ g$^{-1}$, respectively). The tephra $n_s(T)$ values are well above this background but most of the glass $n_s(T)$ values should be regarded as upper limits. The uncertainty in $n_s(T)$ is shown as error bars for a subset of data points (of NUO$_{teph}$ and LIP$_{glass}$, TUN$_{glass}$, AST$_{glass}$, NUO$_{glass}$, LAC$_{glass}$, ETN$_{glass}$, KIL$_{glass}$) and omitted from remaining data points for clarity, but is typical of all samples studied. Sample codes are listed in Table 1.

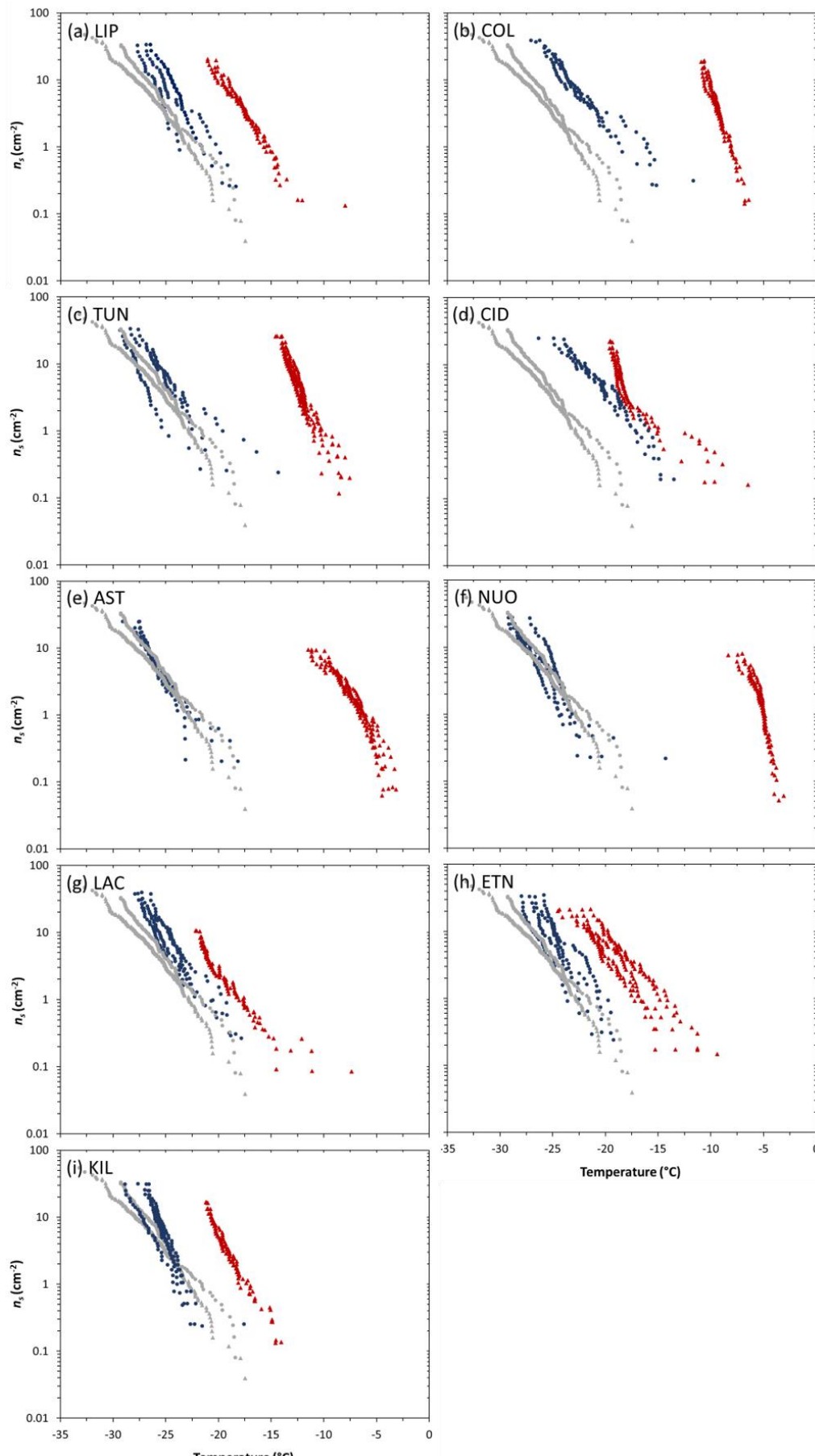

**Figure 3.** Ice nucleation active site density ($n_s$) as a function of temperature for 1 wt.% suspensions of tephra or glass in water. Each plot shows replicates of a compositionally analogous pair of tephra (red triangles) and glass (blue circles). The grey curves with triangle and circle symbols represent the detection limit for $n_s(T)$ based on the background water runs accompanying the tephra and glass experiments, respectively (see Fig. 2 caption for details). Sample codes are listed in Table 1.

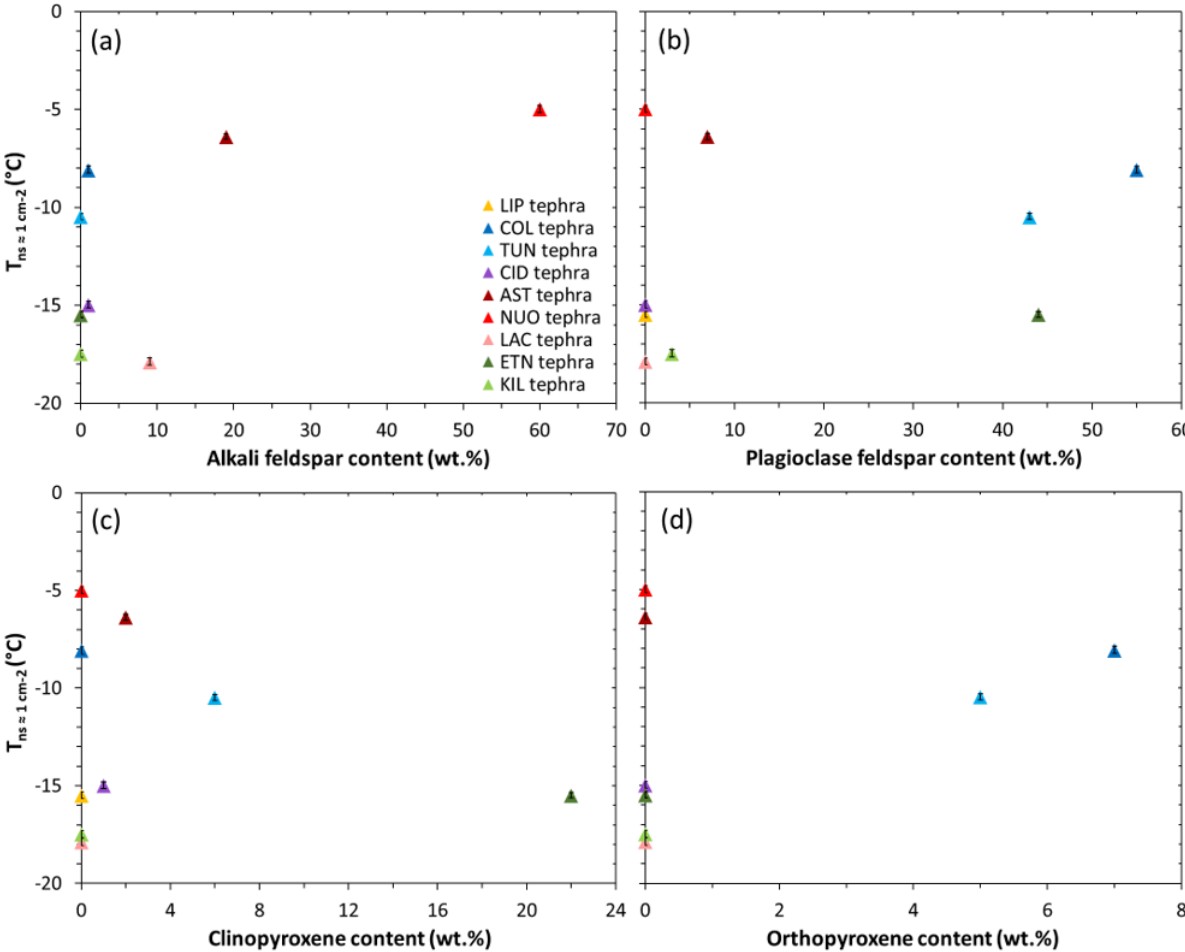

**Figure 4.** The INA ($T_{n_s \approx 1~cm^{-2}}$) of the tephra versus their content of (a) alkali feldspar, (b) plagioclase feldspar, (c) clinopyroxene, and (d) orthopyroxene. Note that minor components below the XRD quantification limit are plotted at 1 wt.%. Ice nucleation experiments were conducted with 1 wt.% suspensions of tephra in water. The uncertainty in $T_{n_s \approx 1~cm^{-2}}$ is shown as error bars (note that these are as small as the data symbols). Sample codes are listed in Table 1.

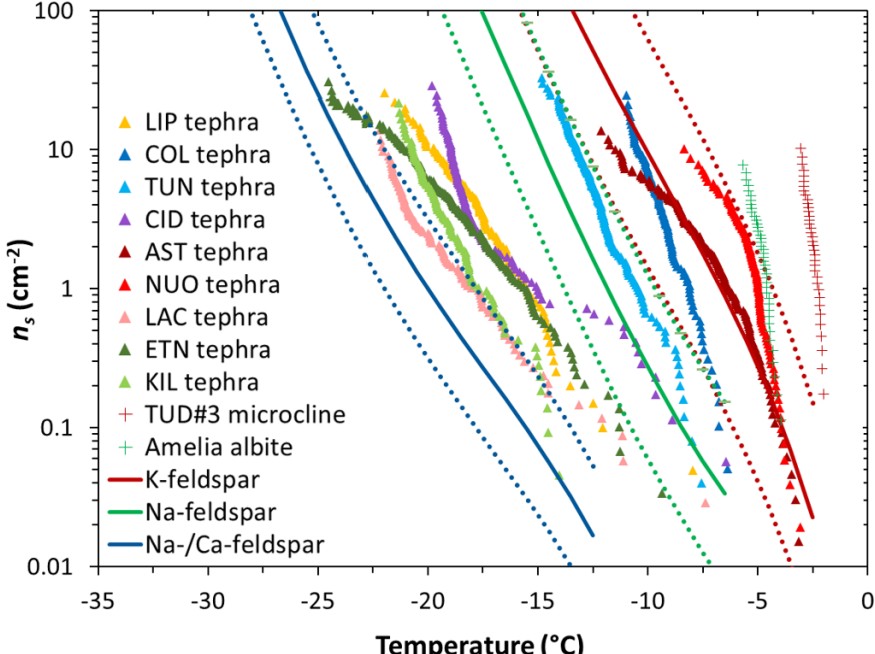

**Figure 5.** Ice nucleation active site density ($n_s$) as a function of temperature for 1 wt.% suspensions of tephra in water. Sample codes are listed in Table 1. The red and green crosses are the $n_s(T)$ values for, respectively, a hyper-active K-feldspar (TUD#3 microcline) and a hyper-active Na-feldspar (Amelia albite) measured by Harrison et al. (2016). The red, green, and blue lines represent parameterisations for, respectively, K-feldspar, Na-feldspar, and Na-/Ca- feldspar of non-pyroclastic origin reported in Harrison et al. (in prep.) from a compilation of literature data, excluding the hyper-active feldspar specimens. The solid lines indicate mean values and the dashed lines indicate lower and upper limits corresponding to the standard deviation of the mean.

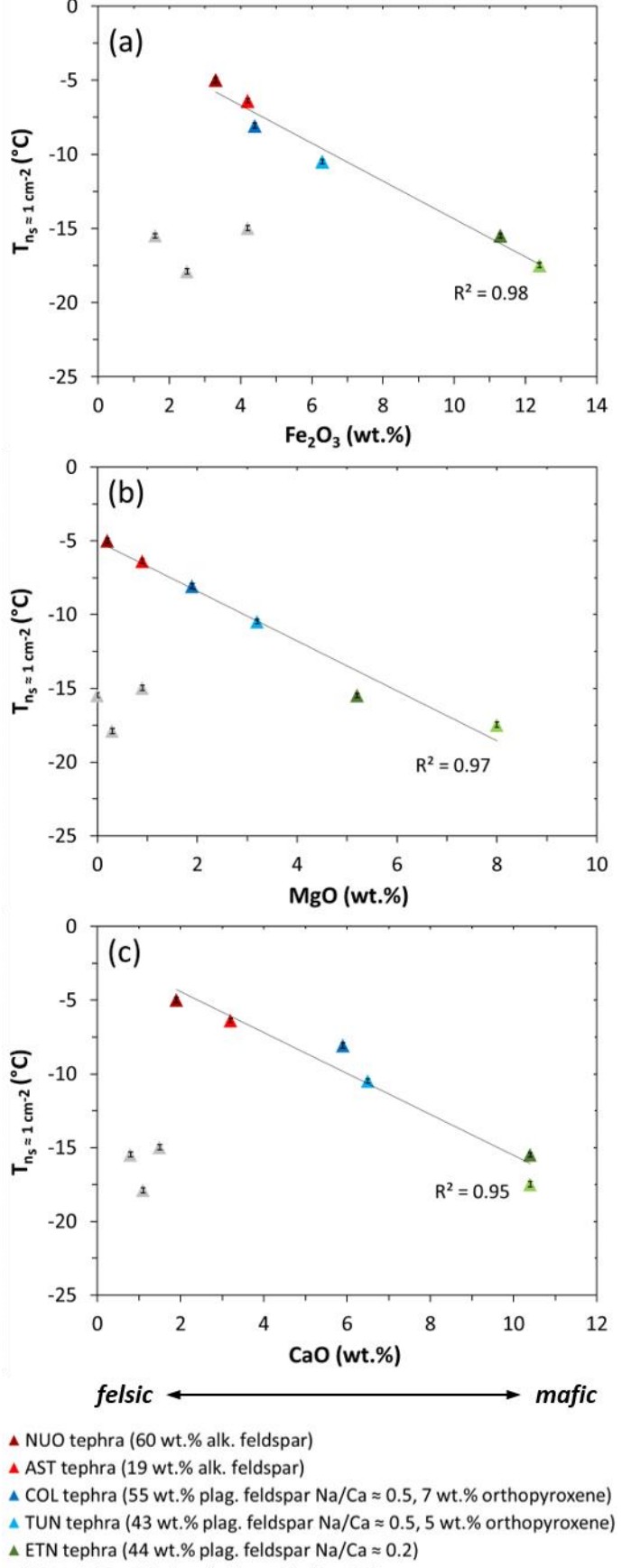

**Figure 6.** The INA ($T_{n_s \approx 1\ cm^{-2}}$) of NUO$_{teph}$, AST$_{teph}$, COL$_{teph}$, TUN$_{teph}$, ETN$_{teph}$, and KIL$_{teph}$ versus their (a) Fe$_2$O$_3$, (b) MgO, and (c) CaO contents. The grey triangles correspond to tephra samples with comparatively low crystallinities (LIP$_{teph}$, CID$_{teph}$, LAC$_{teph}$) which are excluded from the trendline. Ice nucleation experiments were conducted with 1 wt.% suspensions of tephra in water. The uncertainty in $T_{n_s \approx 1\ cm^{-2}}$ is shown as error bars (note that these are as small as the data symbols). Sample codes are listed in Table 1.