# Peer review of "The importance of crystalline phases in ice nucleation by volcanic ash"

_Atmospheric Chemistry and Physics, 2018_

## Referee Comment (RC1) · Anonymous Referee #1 · 1 Feb 2019

SUMMARY

In this work, Maters et al. conducted immersion-freezing ice nucleation experiments on 9 separate samples of volcanic ash. The ashes were chosen to span a range of chemical compositions, crystallinities, and mineralogies. Central to this paper, the authors adapted a technique from volcanology to probe the underlying cause of volcanic ash ice nucleation. In this technique, the ash samples are heated to 1400-1600 C to melt the mineral components of the ash samples. The heated samples were then allowed to cool to room temperature, where the hot ash quenched into a glass and impeded mineral nucleation and growth. The untreated and heated/quenched samples are defined as tephra and glass, respectively. The authors found that, despite being almost identical in chemical compositions, the tephra and glass had vastly different ice

nucleation abilities. Specifically, the tephra always nucleated ice more efficiently than the glass.

Overall, this paper is well written, the techniques used in the paper are well established, and the results are well supported. Furthermore, this work is well within the scope of ACP, and presents novel data to the field of ice nucleation. This reviewer, however, had several qualms with the interpretation of the results. As a general comment, the results from this manuscript represent an incredibly interesting and complex data set. There is a lot of thought-provoking behavior to unpackage, and the reviewer feels like the authors have not fully discussed the results–perhaps because they were beyond their original hypotheses. Instead, the reviewer feels like the authors have jumped to qualitative conclusions–many of which are poorly supported by their experiments. In addition, the authors do a poor job of putting their work in the context of previous works outside of their laboratory. Those comments are outlined below in the "Major Comments" section.

MAJOR COMMENTS

The title of the paper emphasizes the importance of crystalline phases. This, while qualitatively true, is poorly supported by the quantitative techniques in this paper. For example, the LIPteph, LIPglass, CIDteph, and CID glass are all devoid of crystalline material (<2%, i.e., below the limit-of-detection of the instrument), yet they all have vastly different ice nucleation abilities. In addition, their "glass" examples generally differ from their "teph" case. These are two of the nine cases, so they are over 20% of the samples. Ultimately, that some ice active mineral components (or at least 1400-1600 C labile components) that are present at less than 2% is an interesting result from this work that needs further highlighting.

Page, 5, Line 26: In this work alone, there is much evidence that Na/Ca-feldspar is not responsible for ice nucleation. For example, the plagioclase parameterizations from Harrison et al. (in prep) are much too low to explain the ice nucleation efficiency of

almost all of the ash samples in this work, regardless of their plagioclase content. Furthermore, while Harrison et al., 2016 have shown that higher Na2O/CaO may imply higher ice nucleation activity for some feldspars, Page 5, Line 40 is direct contrast to previous studies that have shown that very pure albite is not that ice nucleation active [e.g., Zolles et al., 2015, Schill et al., 2015, Welti et al., 2019]. Alternatively, while there is less literature evidence that orthopyroxene may be the responsible agent, it seems like a much more feasible choice here. It is interesting to the reviewer that the authors then suggest in the conclusion that intermediate to felsic alkaline magmas may then have a higher propensity to contain ice-active ash in their eruptions.

Page 6, Line 1. The electron microprobe studies are not described in the text or in the supplemental. I see from Text S1 that the spot size is ∼10 um; however, it would be useful to know some quantitative limits on this technique as well as how many spot sizes per sample were looked at.

Page 6, Line 9. I am confused by this paragraph. The authors spend a great deal of time describing why LACteph does not have perthitic intergrowth microtexture, but then end the discussion by stating that NUOteph and ASTteph also don't have perthitic intergrowth microtexture. This seems like a logical fallacy to me. A similar sentiment is felt for the section on the anti-rapakivi texture. Was it not observed in the LACteph sample? Finally, how are all of these these surfaces susceptible to changes upon milling with a zirconia ball and vial?

MINOR/TECHNICAL COMMENTS

Page 1, Line 10: This abbreviation seems slightly confusing in the context of ice nucleation, since Vali et al. 2014 have proposed the acronym INE as "ice nucleating entity." I suggest you change INE to something like INeff"

Page 1, Line 15: "warmer" instead of higher?

Page 1, Line 20: The word "categorically" seems excessive here.

Page 1, Line 28: The sentence that starts "Ice formation" is unnecessarily long. I would suggest splitting into at least two sentences. A natural break in theme occurs at "as well as."

Page 1, Line 36: This sentence seems to be missing a comma and coordinating conjunction after diameter. It is the volcanic ash that is usually dominated, not the diameter that is usually dominated.

Page 1, Line 36: This paragraph seems incredibly weak. Part of the problem is that many statements are weakly lumped into "and references therein." For example, there is little mention of previous work on volcanic ash. For something like dust with hundreds of studies converging on a typical behavior, this could be appropriate–however; there are few previous experiments on volcanic ash, and each of them add holistically to the story presented here.

Page 2, Line 14: I would prefer that this sentence, if left here, explains how you "improv[ed] the understanding" instead of just simply stating that it will be done.

Page 2, Line 22: Since this is an atmospheric chemistry and physics journal, and not a geology journal, it would be helpful for readers of this journal to have the melting point range of each of the minerals in Table 2 compiled for them. That would greatly help them interpret the results of this study and perhaps elucidate why some samples retain some ice nucleation activity after the melt/quench cycle.

Page 3, Line 28 and Line 37: These equations need commas after them, since you have another clause after them (starting with "where").

Page 4, Line 10: I see why these arguments are included in this section, but they seem out of place. For example, the glassy organic particles all nucleated ice in the "deposition" mode and were certainly not immersed in water droplets at or above water saturation. I would suggest the first two sentences of this paragraph be qualified or removed.

[Figure]
**Interactive comment**

Page 4, Line 16: As the figure looks now, there does not seem to be significant overlap with the markers and the grayed out region. Perhaps adding error bars to all curves would make that more clear? Or put error bars on the points in Figure 3?

Page 6, Line 28: Or, potentially, mafic magmas with high orthopyroxene?

Page 7, Line 20: There should be a comma between orthopyroxene and which.

Page 7, Line 37: But–isn't the ash you collected from volcanic plumes where high concentrations of acidic gases likely already interacted with the ash?

Figure 1 Legend: Is LEI in Figure 1 KIL everywhere else?

Figure 2, 3, and 5. The symbols in these figures are unreasonably small, especially in print form. This makes it relatively difficult to see the different between some of the samples that have similar marker colors (e.g., green, red, light blue.)

ADDITIONAL REFERENCES

Zolles T. et al., [2015], J. Phys. Chem. A, 119 (11), 2692-2700, DOI: 10.1021/jp509839x

Welti, A., Lohmann, U., and Kanji, Z.A., [2019], Atmos. Chem. Phys. Discuss., DOI: 10.5194/acp-2018-1271

---

## Referee Comment (RC2) · Anonymous Referee #2 · 17 Feb 2019

General Comments:

The manuscript of Maters et al. describes immersion-mode ice nucleation experiments conducted on volcanic ash samples from several different volcanoes with variable magma compositions. The ice nucleation activity is compared between tephras (containing both glass and minerals) and the pure glass equivalents of these tephras. This is a novel approach to exploring the role of crystalline vs. amorphous phases in volcanic ash as ice nuclei, and the study is appropriate to ACP. The difficult nature of immersion-mode experiments and the limited knowledge of volcanic ash nucleation activity makes this study an important contribution to the state of knowledge. The manuscript is well-constructed and well-written. I think it will be suitable for publication following some minor revisions and further interpretation. Some more specific com-

ments are provided below.

Specific Comments:

Throughout the manuscript you refer to the vertical eruption "plume", and the laterally dispersed "cloud", as defined on line 27 of the introduction. I understand that these terms are not explicitly set by anyone, but physically, it is more correct to call the vertical part the "column" and the laterally spreading part the "plume." Calling the plume a "cloud" is not technically correct and can cause some confusion amongst atmospheric scientists.

Why not refer to the ice nucleation activity (INA) instead of the ice-nucleating effectiveness (INE). INE is already used for other descriptors.

In the Materials and Methods section, more information on the preparation of the tephra samples is required. Were accidental lithics removed? Were the samples rinsed to remove adsorbed salts? This second question relates to the statement on page 7, line 37. Were they altered in any way following eruption? Weathering of the glass post-deposition may introduce small amounts of clay minerals into the samples that are not in high enough quantity to be detected by XRD.

Following milling of the tephras, was a grain size distribution analysis performed to insure the sizes of particles were consistent between samples? Although surface area was measured, the size distribution of the particles may also affect the ice nucleation (i.e., more smaller particles will increase SA compared to fewer, larger particles). Although the milling procedure should effectively homogenize the size distribution, checking this would strengthen the reliability of the results.

Although they used deposition-mode experiments, and not immersion-mode, the recent study of Kiselev et al. (2017) examined variations in ice nucleation due to defects in the crystal structure of K-feldspars. It may be worthwhile to look at this study for additional information on your interpretations. Specifically, it may be worth considering

not only the presence of particular minerals, but the crystal shapes and surface attributes of these minerals, as they may also affect the likelihood of ice nucleation. This point is partly discussed on page 6, line 5, regarding the study of Whale et al. (2017), but should be explored further. I have included the Kiselev reference below.

Additionally, it is mentioned on page 6, line 18, that any relevant mineral textures are difficult to resolve in powdered samples, but these samples can be easily examined in backscattered SEM mode to check for any notable mineral textures. Without knowing the size of the grains, it is difficult to say for sure.

Technical Corrections:

Line 19 of the abstract: delete the word "partly"

The sentence beginning on page 2, line 35 just sounds awkward and should be rephrased.

The third and fourth paragraphs in section 2.1 are not materials or methods and should be placed somewhere else, perhaps the introduction?

Page 7, line 21, tephra should be plural.

Can you discuss further the characteristics of pyroxene minerals that might influence their ice nucleation abilities?

I don't quite understand the point of plotting both the tephras and glasses in Figure 1a, since they directly overlap in most cases.

When reporting the INE (Tns $\sim$ 1 cm -2) throughout the text, why is the 1 so small...is it subscripted?

Please state in the caption of Table S1 that these are XRF measurements.

Please state in the caption of Table S2 that these are XRF measurements.

Is the "Text S1" mislabeled? I think the supplementary tables are not currently labeled

correctly. Should they be: Table S1 (XRF measurements of bulk samples); Table S2 (Electron microprobe measurements of tephra glasses; Table S3 (Electron microprobe measurements of feldspars).

Figures 4, 6, S1, and S2 need to include error bars or a statement of the errors in the captions.

Additional References:

Kiselev, A.; Bachmann, F.; Pedevilla, P.; Cox, S.J.; Michaelides, A.; Gerthsen, D.; Leisner, T. Active sites in heterogeneous ice nucleation—The example of K-rich feldspars. Science 2017, 355, 367–371.

---

## Author Comment (AC1) · 24 Mar 2019

**Authors' response to Reviewers' comments on "The importance of crystalline phases in ice nucleation by volcanic ash" (acp-2018-1326)**

We thank the two Reviewers for their helpful evaluation of our manuscript. We have been able to respond to all comments and provide a carefully revised manuscript. The comments in italic and our responses in normal type are given below. Examples of relevant text (existing or new) are presented in grey highlight between quotation marks. The line numbers that we refer to correspond to those in the revised manuscript.

**REVIEWER #1**

**Major Comments**

The title of the paper emphasizes the importance of crystalline phases. This, while qualitatively true, is poorly supported by the quantitative techniques in this paper. For example, the LIPteph, LIPglass, CIDteph, and CIDglass are all devoid of crystalline material (<2%, i.e., below the limit-of-detection of the instrument), yet they all have vastly different ice nucleation abilities. In addition, their "glass" examples generally differ from their "teph" case. These are two of the nine cases, so they are over 20% of the samples. Ultimately, that some ice active mineral components (or at least 1400-1600 C labile components) that are present at less than 2% is an interesting result from this work that needs further highlighting.

We agree with the Reviewer that differences in INA between  $LIP_{teph}$ ,  $LIP_{glass}$ ,  $CID_{teph}$ , and  $CID_{glass}$ , which all have <2 wt.% crystalline material, should be addressed given our emphasis of the importance of crystalline phases.

We acknowledge already on Page 4 Lines 8-10 that "While the LIPteph and CIDteph crystallinities cannot be quantified below the ~2 wt.% limit of the technique, this does not rule out the possibility of smaller amounts of crystals and/or nanoscale crystallites being present in these samples." Table 2 lists crystalline phases that might occur as minor components below quantification in the tephra, including Fe(-Ti) oxide in LIPteph, and alkali feldspar, clinopyroxene, and Fe(-Ti) oxide in CIDteph. These minor components could be why the two dominantly glassy tephra samples show different INA and are still more ice-active than their counterpart glass samples (Figure 3a,d). We have inserted text in the discussion to highlight this:

Page 5 Lines 33-35: "Even the dominantly glassy LIPteph and CIDteph (<2 wt.% crystallinity) display higher INA than their counterpart LIPglass and CIDglass, which could reflect the influence of minor crystalline components (below quantification by XRD; Table 2) in these tephra on their ability to nucleate ice."

The observation that  $CID_{teph}$  is only slightly more ice-active than  $CID_{glass}$  (Figure 3d) may, as the Reviewer suggests, reflect some ice-active mineral components also present in amounts <2 wt.% in this glass sample following melting, homogenising, and quenching. This possibility is supported by the observation that  $CID_{glass}$  nucleates ice at warmer temperatures than the background water and the other glass samples (Figure 2b,d). We have inserted text in the discussion to highlight this also:

Page 8 Lines 26-28: "It is possible that CIDglass and COLglass contain very small amounts of crystals below detection by XRD that survived melting or formed during quenching, which could explain why these glasses stand out in their ability to nucleate ice."

Overall, we think that our findings, both quantitative and qualitative, support the title of the paper.

Page, 5, Line 26: In this work alone, there is much evidence that Na/Ca-feldspar is not responsible for ice nucleation. For example, the plagioclase parameterizations from Harrison et al. (in prep) are much too low to explain the ice nucleation efficiency of almost all of the ash samples in this work, regardless of their plagioclase content. Furthermore, while Harrison et al., 2016 have shown that higher Na2O/CaO may imply higher ice nucleation activity for some feldspars, Page 5, Line 40 is direct contrast to previous studies that have shown that very pure albite is not that ice nucleation active [e.g., Zolles et al., 2015, Schill et al., 2015,

Welti et al., 2019]. Alternatively, while there is less literature evidence that orthopyroxene may be the responsible agent, it seems like a much more feasible choice here. It is interesting to the reviewer that the authors then suggest in the conclusion that intermediate to felsic alkaline magmas may then have a higher propensity to contain ice-active ash in their eruptions.

We acknowledge that we can improve the presentation of our arguments in this section. The Reviewer is correct that the high INA of tephra samples containing plagioclase feldspar (Page 6 Lines 34-35) "are inconsistent with the relatively low INA of Na-/Ca-feldspar reported in the literature (Fig. 5)." However, evidence of exceptionally ice-active feldspars including a hyper-active Na-feldspar (Amelia albite; characterised by the plagioclase structure) has been presented by Harrison et al. (2016), which we mention and now plot in Figure 5 to illustrate that highly ice-active plagioclase feldspar exists in nature. Even excluding this hyper-active albite, and contrary to the Reviewer's remark that "*albite is not that ice nucleation active*", a new Na-feldspar parameterisation compiled from literature data (Harrison et al., in prep.), which we have also added to Figure 5, falls much closer to the high INA shown by COLteph and TUNteph (Fig. R1 below). We have revised the text as follows:

Page 6 Lines 35-40/Page 7 Lines 1-4: "This may point to the presence in COLteph and TUNteph of ice-active plagioclase feldspar characterised by an INA closer to the Na-feldspar (albite) parameterisation, or potentially more akin to the hyper-active feldspars measured by Harrison et al. (2016; Fig. 5). It is not clear why these hyper-active samples (Amelia albite and TUD#3 microcline) have a much greater INA relative to the majority of feldspars tested (Harrison et al., 2016; Peckhaus et al., 2016), but such wide variability may relate to the specific mechanisms and/or conditions of formation and subsequent processing of individual samples (Welti et al., 2019), and it might be that plagioclase feldspar in COLteph and TUNteph was produced in a way that gives rise to enhanced activity."

Together, the three parameterisations and two hyper-active samples clearly demonstrate the wide variability in INA of feldspar minerals investigated to date. In light of this variability, we maintain our suggestion that highly ice-active plagioclase feldspar might occur in COLteph and TUNteph.

**Figure R1.** Ice nucleation active site density ( $n_s$ ) as a function of temperature for 1 wt.% suspensions of tephra in water. The red and green crosses are the  $n_s(T)$  values for, respectively, a hyper-active K-feldspar (TUD#3 microcline) and a hyper-active Na-feldspar (Amelia albite) measured by Harrison et al. (2016). The red, green, and blue lines represent parameterisations for, respectively, K-feldspar, Na-feldspar, and Na-/Ca-feldspar reported in Harrison et al. (in prep.) from a compilation of literature data, excluding the hyper-active feldspar specimens. The solid lines indicate mean values and the dashed lines indicate lower and upper limits corresponding to the standard deviation of the mean.

We understand the Reviewer's sentiment "that orthopyroxene may be the responsible agent" and we acknowledge (Page 7 Lines 4-5) that the high INA of COLteph and TUNteph alternatively "may relate to the influence of some other mineral component such as orthopyroxene" on ice nucleation by these samples. However, as the Reviewer points out, there is limited literature on ice nucleation by pyroxenes, and so we conclude (Page 8 Lines 11-13) that "we cannot rule out a potential influence of pyroxene on ice nucleation by volcanic ash" and "additional research is needed to quantify the INA of a range of pyroxene minerals, and probe the nature of their ice-nucleating properties, in order to better inform this assessment." We have additionally highlighted a very recent study on ice nucleation by volcanic ash and pyroxene in our discussion:

Page 6 Lines 22-23: "More recently, Jahn et al. (2019) proposed that feldspar and pyroxene were responsible for ice nucleation by ash from Soufrière Hills, Fuego, and Santiaguito volcanoes."

Page 8 Lines 6-11: "More recently, Jahn et al. (2019) measured the INA of a clinopyroxene specimen (freezing from -8 to -24 °C), citing its behaviour to explain the INA of three pyroxene-containing volcanic ash samples. However, XRD analysis indicated that this specimen comprising diopside-augite also contained ~5 wt.% feldspar, which might have influenced the INA observed. In any case, it should be emphasised that a single mineral specimen might not provide a good representation of the INA of a given mineral type, as shown by studies on ice nucleation by feldspar and quartz (Harrison et al., 2016; Whale et al., 2017; Harrison et al., in prep.)."

Our own measurements of a range of pyroxene samples (wollastonite, diopside, augite, enstatite, hypersthene) do not show these minerals to be especially ice-active (Maters et al., in prep.). Therefore, we do not agree that orthopyroxene "*seems like a much more feasible choice*" to explain the high INA of COLteph and TUNteph in our study. However, we recognise that some content in our original manuscript conveyed a preference for the hypothesis of ice-active plagioclase feldspar over that of ice-active orthopyroxene. In light of the Reviewer's comments, and given that we currently lack enough information to conclude definitively which may be the most ice-active mineral(s) in the studied tephra (specifically COLteph and TUNteph), we have modified this content so as not to favour either hypothesis:

Page 1 Lines 17-19: "There is evidence of a potential indirect relationship between chemical composition and ash INA, whereby a magma of felsic to intermediate composition may generate ash containing highly ice-active feldspar or pyroxene minerals."

Page 6 Lines 26-28: "[...] while the next most ice-active COLteph and TUNteph are characterised by an abundance of plagioclase feldspar (55 and 43 wt.%, respectively; Fig. 4b) and lesser amounts of orthopyroxene (7 and 5 wt.%, respectively, Fig. 4d)."

Page 8 Lines 33-35: "[...] the  $T_{n_s \approx 1 \text{ cm}^{-2}}$  decreases with increasing Fe2O3, MgO, and CaO contents for NUOteph, ASTteph, COLteph, TUNteph, ETNteph, and KILteph (Fig. 6), in an order consistent with interpretations relating to their feldspar contents and chemistries and/or a potential effect of orthopyroxene in two of these samples (see discussion Sect. 4.2)."

Page 9 Lines 17-20: "[...] we speculate that highly ice-active ash particles might be erupted by volcanoes with intermediate to felsic alkaline magmas giving rise to feldspar crystals featuring overgrowth textures (e.g., Astbury et al., 2016; 2018) or potentially pyroxene crystals with high INA for reasons yet unknown (e.g., Jahn et al., 2019)."

Figure 6: We have inserted the orthopyroxene content of  $COL_{teph}$  and  $TUN_{teph}$  in the legend.

Lastly, the Reviewer implies that the possibility that (ortho)pyroxene is highly ice-active contradicts our concluding speculation (Page 9 Lines 17-20) that intermediate to felsic alkaline magmas might give rise to highly ice-active ash, presumably because pyroxene can also form in mafic magmas (Figure 1a). However, this

speculation is borne from observations that the most ice-active samples (COLteph, TUNteph, NUOteph, ASTteph) are those originating from intermediate to felsic magmas (Figure 6), and is not fundamentally affected by whether it is the (ortho)pyroxene, plagioclase feldspar, or alkali feldspar driving ice nucleation by these samples. In contrast, no empirical evidence leads us to speculate that mafic magmas might give rise to highly ice-active ash, as the much less ice-active samples (ETNteph, KILteph) containing (clino)pyroxene and/or plagioclase feldspar are those originating from mafic magma (Figure 6). We infer (Page 7 Lines 8-9) that even when tephra samples contain common minerals, "differences [in INA] might relate to the specific chemistry of the mineral phases present in the tephra (Zolles et al., 2015; Welti et al., 2019)," which likely varies depending on factors including magma composition. We thus stand by our original suggestion that intermediate to felsic magmas might erupt highly ice-active crystalline ash.

Page 6, Line 1. The electron microprobe studies are not described in the text or in the supplemental. I see from Text S1 that the spot size is ~10 um; however, it would be useful to know some quantitative limits on this technique as well as how many spot sizes per sample were looked at.

The electron microprobe analysis is described in the Supplementary Material, specifically in Text S1, which is referred to by the Reviewer. We have added details regarding the number of spots analysed as well as quantitative limits on this technique:

Supplement Page 3 Lines 2-5: "A 10  $\mu$ m focused beam was used at an accelerating voltage of 15 keV and a current of 5 nA to analyse at least five points for each crystalline phase in the tephra samples. Elemental detection limits in parts per million are as follows: Si - 786, Al - 655, Fe - 1573, Mg - 501, Ca - 747, Na - 973, K - 711, Ti - 894, Mn - 1401, P - 568, Cr - 1286, S - 767, Cl - 955."

Page 6, Line 9. I am confused by this paragraph. The authors spend a great deal of time describing why LACteph does not have perthitic intergrowth microtexture, but then end the discussion by stating that NUOteph and ASTteph also don't have perthitic intergrowth microtexture. This seems like a logical fallacy to me. A similar sentiment is felt for the section on the anti-rapakivi texture. Was it not observed in the LACteph sample? Finally, how are all of these these surfaces susceptible to changes upon milling with a zirconia ball and vial?

We have revised the text in this paragraph to improve our reasoning and have added reference to a very recent study of ice nucleation by various K-feldspars (Welti et al., 2019):

Page 7 Lines 22-39: "However, perthite in alkali feldspar develops in metamorphic and plutonic contexts during slow cooling at temperatures <700 °C (Parsons, 2010), and is generally not expected in volcanic ash which cools rapidly from magmatic down to ambient temperatures during explosive eruption (Parsons et al., 2015). An absence of perthitic microtexture is consistent with the low INA of LACteph in spite of its alkali feldspar content (9 wt.%). This is supported by evidence that the alkali feldspar mineral sanidine sourced from the same geological setting as LACteph (Eifel volcanic field) lacks perthitic texture and exhibits a poor ability to nucleate ice (Whale et al., 2017). A recent study similarly found volcanic sanidine from Germany to be the least iceactive among the alkali feldspar samples tested (Welti et al., 2019). In contrast, an absence of perthitic microtexture is inconsistent with the high INA of NUOteph and ASTteph, and perhaps some other textural feature underlies these samples' ability to nucleate ice as effectively as alkali feldspar of non-pyroclastic origin (Fig. 5). Pyroclastic material from both the 1538 Monte Nuovo eruption (i.e., the origin of NUOteph) and the ~4 ka Astroni eruption (i.e., the origin of ASTteph) has been found to contain anti-rapakivi overgrowth microtexture characterised by plagioclase feldspar cores rimmed by alkali feldspar (D'Oriano et al., 2005; Astbury et al., 2016; 2018). We are not aware of any studies reporting similar textures in pyroclastic material from the 12.9 ka Laacher See eruption (i.e., the origin of  $LAC_{teph}$ ). Such textures are challenging to resolve optically in powdered samples including the tephra studied here. Further optical and microanalytical (Scanning- and Transmission Electron Microscopy) observations (e.g., Whale et al., 2017; Holden et al., 2019) will be needed to explore whether the boundary between Na- and K-rich regions in anti-rapakivi microtexture may give rise to nanoscale topography that induces effects analogous to perthitic microtexture in promoting ice nucleation."

Lastly, the Reviewer queries how such textures might be affected by milling with a zirconia ball and vial. Tephra surfaces are fractured as coarse particles are crushed down to finer particles. However, as we note on Page 7 Lines 17-21, the ability of textures such as perthite to promote ice nucleation is thought to relate to nanoscale topographical features at the boundary of Na- and K-rich phases (Whale et al., 2017; Holden et al., 2019). These nanoscale features would be preserved in milled particles ranging from tens to hundreds of nanometres to several micrometres in diameter.

**Minor/Technical Comments**

Page 1, Line 10: This abbreviation seems slightly confusing in the context of ice nucleation, since Vali et al. 2014 have proposed the acronym INE as "ice nucleating entity." I suggest you change INE to something like INeff"

We have changed "INE" for "ice-nucleating effectiveness" to "INA" for "ice-nucleating activity" throughout the manuscript and Supplementary Material.

Page 1, Line 15: "warmer" instead of higher?

We have substituted "warmer" into this line.

Page 1, Line 20: The word "categorically" seems excessive here.

We have removed "categorically" from this line.

Page 1, Line 28: The sentence that starts "Ice formation" is unnecessarily long. I would suggest splitting into at least two sentences. A natural break in theme occurs at "as well as."

We have split this sentence into two.

Page 1, Line 36: This sentence seems to be missing a comma and coordinating conjunction after diameter. It is the volcanic ash that is usually dominated, not the diameter that is usually dominated.

We have revised this sentence as follows:

Page 1 Lines 37-39: "By definition, volcanic ash consists of pyroclastic particles <2 mm in diameter, and is comprised of aluminosilicate glass as well as aluminosilicate and/or Fe(-Ti) oxide minerals (Heiken and Wohletz, 1992)."

Page 1, Line 36: This paragraph seems incredibly weak. Part of the problem is that many statements are weakly lumped into "and references therein." For example, there is little mention of previous work on volcanic ash. For something like dust with hundreds of studies converging on a typical behavior, this could be appropriate—however; there are few previous experiments on volcanic ash, and each of them add holistically to the story presented here.

We have added content in this paragraph relating to previous studies on volcanic ash ice nucleation:

Page 2 Lines 17-20: "In immersion freezing experiments, Soufrière Hills ash has been found to range from inactive to highly active in nucleating ice, with the discrepancy inferred to relate to differences in ash composition and sample preparation methods (Schill et al., 2015; Mangan et al., 2017; Jahn et al., 2019)."

Page 2 Lines 24-33: "There is increasing evidence that similar factors may influence ice nucleation by volcanic ash. Kulkarni et al. (2015) argued that the presence of amorphous material reduced the INA of Eyjafjallajökull ash compared to Arizona test dust, based on the notion that crystalline structures provide preferred

configurations for water molecules to bind at the particle surface (Pruppacher and Klett, 2010). Schill et al. (2015) proposed that, aside from amorphous versus crystalline content, differences in mineralogy could explain the INA of ash from Soufrière Hills, Fuego, and Taupo volcanoes. Recently, Jahn et al. (2019) suggested that feldspar and pyroxene minerals were responsible for the high INA of Soufrière Hills, Fuego, and Santiaguito ash samples. Genareau et al. (2018) conversely noted a broad trend between chemical composition and ice nucleation, with the INA of five ash samples increasing with K2O content and decreasing with MnO content."

**Page 2, Line 14: I would prefer that this sentence, if left here, explains how you "improv[ed] the understanding" instead of just simply stating that it will be done.**

We have revised this sentence as follows:

Page 2 Lines 41-42/Page 3 Lines 1-2: "By finding that crystalline phases promote ash ice nucleation and that magma composition may exert an indirect effect via its influence on ash mineralogy, we contribute to an improved understanding of the potential for airborne ash from different eruptions to impact ice formation above the volcanic vent and/or once dispersed in the ambient atmosphere."

**Page 2, Line 22: Since this is an atmospheric chemistry and physics journal, and not a geology journal, it would be helpful for readers of this journal to have the melting point range of each of the minerals in Table 2 compiled for them. That would greatly help them interpret the results of this study and perhaps elucidate why some samples retain some ice nucleation activity after the melt/quench cycle.**

The melting points of minerals in a pure state are different from the melting points of minerals in a heterogeneous mixture such as volcanic tephra. Therefore, it would not be meaningful to report "*the melting point range of each of the minerals in Table 2*", since they would not be reflective of the actual temperature at which the tephra components melted (eutectic melting) during the process of generating the glass samples.

However, the Reviewer makes a good point in implying that the observation that "*some samples retain some ice nucleation activity after the melt/quench cycle*" might be indicative of small amounts of crystals being present in a couple of the glass samples following melting and quenching. We have inserted text to highlight this possibility in explaining the higher INA of CIDglass and COLglass relative to the other glasses studied:

Page 8 Lines 26-28: "It is possible that CIDglass and COLglass contain small amounts of crystals below detection by XRD that survived melting or formed during quenching, which could explain why these glasses stand out in their ability to nucleate ice."

**Page 3, Line 28 and Line 37: These equations need commas after them, since you have another clause after them (starting with "where").**

We have inserted commas after these equations.

Page 4, Line 10: I see why these arguments are included in this section, but they seem out of place. For example, the glassy organic particles all nucleated ice in the "deposition" mode and were certainly not immersed in water droplets at or above water saturation. I would suggest the first two sentences of this paragraph be qualified or removed.

We have removed reference here to ice nucleation by glassy organic particles.

Page 4, Line 16: As the figure looks now, there does not seem to be significant overlap with the markers and the grayed out region. Perhaps adding error bars to all curves would make that more clear? Or put error bars on the points in Figure 3?

There is considerable overlap in terms of the temperature range in which freezing of the background water and the glass sample suspensions occurs (not necessarily overlap of individual data points). We have revised the text to clarify this and, as the Reviewer suggests, we have added error bars to the  $n_s(T)$  curves of the glass samples in the region of the background water in Figure 2d:

Page 5 Lines 12-15: "For the glass samples in contrast, there is significant overlap of temperatures at which their  $f_{ice}(T)$  curves and those of the background water fall, spanning a range from -18 °C to -35 °C (Fig. 2b). As illustrated by the overlap of error bars in their  $n_s(T)$  curves (Fig. 2d), this prevents attribution of the observed freezing to ice nucleation by glass particles and comparison of individual glass activities."

**Page 6, Line 28: Or, potentially, mafic magmas with high orthopyroxene?**

Please see our detailed response above to the Reviewer's second major comment, which relates to the same idea. We agree and acknowledge that pyroxene minerals, as well as feldspar minerals, could be responsible for ice nucleation by volcanic ash. However, based on experimental observations regarding the most ice-active tephra samples (COLteph, TUNteph, NUOteph, ASTteph), we maintain our original speculation in the conclusion that intermediate to felsic alkaline magmas might give rise to crystalline ash that is highly ice-active (Figure 6).

**Page 7, Line 20: There should be a comma between orthopyroxene and which.**

We have inserted a comma here.

**Page 7, Line 37: But-isn't the ash you collected from volcanic plumes where high concentrations of acidic gases likely already interacted with the ash?**

The tephra samples correspond to either ash (COLteph, TUNteph, ASTteph) or pumice (LIPteph, CIDteph, NUOteph, LACteph, ETNteph, KILteph), which to varying extents, likely already interacted with acidic gases and condensates while airborne and may have experienced leaching by water once deposited. However, as we note on Page 3 Lines 11-12: "All samples were crushed to fine powders in a ball mill using a zirconia ceramic ball and vial to ensure consistent treatment of the tephra and glass materials prior to ice nucleation experiments." We have inserted additional text to clarify this in the Materials and Methods:

Page 3 Lines 12-18: "This also reduced the influence of chemically-altered surfaces resulting from tephra interaction with gases and/or liquids post-eruption (e.g., Delmelle et al., 2007), by exposing fresh surfaces with chemical and mineralogical properties reflective of the source magma and any entrained lithic material. This allowed us to address the study objective of assessing specifically the role of chemical composition, crystallinity, and mineralogy on ice nucleation by volcanic ash. Aside from crushing, the samples were not processed by rinsing with water or otherwise, to avoid further alteration of these materials on short time scales (e.g., exposure to water is known to change the INA of some minerals; Harrison et al., 2016; Kumar et al., 2018)."

This is why we acknowledge on Page 9 Lines 28-41 of the Conclusion that - having studied the importance of these primary properties (chemical composition, crystallinity, mineralogy) in ice nucleation by volcanic ash - the next step is to investigate how ash surface 'aging' in the plume and the atmosphere may influence the ash INA. Such further investigations are in progress in our laboratory (e.g., Maters et al., 2019).

**Figure 1 Legend: Is LEI in Figure 1 KIL everywhere else?**

Yes; we have replaced "LEI" in the Figure 1 legend with "KIL" as elsewhere

Figure 2, 3, and 5. The symbols in these figures are unreasonably small, especially in print form. This makes it relatively difficult to see the different between some of the samples that have similar marker colors (e.g., green, red, light blue.)

We have amended this; enlarging the symbols in these figures to make it easier to see different sample colours, yet still small enough to distinguish individual sample curves (avoid extensive overlap of closely plotted data points).

**REVIEWER #2**

**Specific Comments**

Throughout the manuscript you refer to the vertical eruption "plume", and the laterally dispersed "cloud", as defined on line 27 of the introduction. I understand that these terms are not explicitly set by anyone, but physically, it is more correct to call the vertical part the "column" and the laterally spreading part the "plume." Calling the plume a "cloud" is not technically correct and can cause some confusion amongst atmospheric scientists.

Unfortunately, these terms are not used systematically in the literature. We have adopted terminology consistent with leading authors in the field, where the vertical component connected to the active volcanic vent is called the eruption "plume" (e.g., Herzog et al., 1998; Delmelle et al., 2007; Ayris et al., 2013; 2014; Hoshyaripour et al., 2012; 2014; Van Eaton et al., 2015). Similarly, we call the laterally dispersed component the eruption "cloud", consistent with literature referring to this more dilute feature downwind of the active volcanic vent (e.g., Rose et al., 1995; 2000; 2006; Durant et al., 2010; 2012; Van Eaton et al., 2015).

Why not refer to the ice nucleation activity (INA) instead of the ice-nucleating effectiveness (INE). INE is already used for other descriptors.

We have changed "INE" for "ice-nucleating effectiveness" to "INA" for "ice-nucleating activity" throughout the manuscript and Supplementary Material.

In the Materials and Methods section, more information on the preparation of the tephra samples is required. Were accidental lithics removed? Were the samples rinsed to remove adsorbed salts? This second question relates to the statement on page 7, line 37. Were they altered in any way following eruption? Weathering of the glass postdeposition may introduce small amounts of clay minerals into the samples that are not in high enough quantity to be detected by XRD.

In short, nothing was done to the tephra samples aside from crushing them in a ball mill. We have added text to explain the reasoning for this in the Materials and Methods:

Page 3 Lines 11-18: "All samples were crushed to fine powders in a ball mill using a zirconia ceramic ball and vial to ensure consistent treatment of the tephra and glass materials prior to ice nucleation experiments. This also reduced the influence of chemically-altered surfaces resulting from tephra interaction with gases and/or liquids post-eruption (e.g., Delmelle et al., 2007), by exposing fresh surfaces with chemical and mineralogical properties reflective of the source magma and any entrained lithic material. This allowed us to address the study objective of assessing specifically the role of chemical composition, crystallinity, and mineralogy in ice nucleation by volcanic ash. Aside from crushing, the samples were not processed by rinsing with water or otherwise, to avoid further alteration of these materials on short time scales (e.g., exposure to water is known to change the INA of some minerals; Harrison et al., 2016; Kumar et al., 2018)."

We have also added text to acknowledge the possibility of soluble salts in the tephra affecting ice nucleation in the Conclusion:

Page 9 Lines 34-39: "Moreover, it has recently been shown that even very low concentrations of soluble salts  $(1 \times 10^4 \text{ M})$  can influence the INA of feldspar minerals (Whale et al., 2018; Kumar et al., 2018), and we cannot exclude the possibility that small amounts of NaCl or KCl formed by prior ash-gas/condensate interactions in our tephra samples reduced their INA. However, given the strong correlations observed between INA and composition of the crystalline tephra samples (Fig. 6), we do not think that a potential influence of soluble salts on freezing temperatures affects the general conclusions of this study."

Lastly, weathering of some of the tephra samples post-deposition could have introduced small amounts of clay minerals below detection by XRD, as the Reviewer suggests. However, clay minerals are not thought to be particularly ice-active (e.g., Atkinson et al., 2013; Augustin-Bauditz et al., 2014), and if present in such small quantities are unlikely to have driven the overall trends observed for the crystalline tephra samples, whose INAs are found to correlate well with their bulk compositional and mineralogical properties (Table 2; Figure 6).

Following milling of the tephras, was a grain size distribution analysis performed to insure the sizes of particles were consistent between samples? Although surface area was measured, the size distribution of the particles may also affect the ice nucleation (i.e., more smaller particles will increase SA compared to fewer, larger particles). Although the milling procedure should effectively homogenize the size distribution, checking this would strengthen the reliability of the results.

The Reviewer raises the idea that grain size distribution differences between samples could affect ice nucleation since more smaller particles provide greater surface area than fewer larger particles. We agree with this and hence, have normalised the ice nucleation data to the total surface area in each experiment (see Equation 2, Page 4 Lines 31-34), calculated from the sample specific surface area (in m2 g-1) determined by nitrogen gas adsorption and the sample mass present in the water droplets (given a 1 wt.% suspension). Expressing the INA of samples in terms of the number of ice nucleation active sites per unit surface area (' $n_s$ '), as done widely in literature studies of heterogeneous ice nucleation (e.g., Connolly et al., 2009; Hoose and Möhler, 2012; Zolles et al., 2015; Harrison et al., 2016; Jahn et al., 2019), implicitly accounts for differences in solid surface area provided by particles of different sizes between samples. Therefore, we do not think that there is an added value in presenting the grain size distribution of the milled samples in this study.

Although they used deposition-mode experiments, and not immersion-mode, the recent study of Kiselev et al. (2017) examined variations in ice nucleation due to defects in the crystal structure of K-feldspars. It may be worthwhile to look at this study for additional information on your interpretations. Specifically, it may be worth considering not only the presence of particular minerals, but the crystal shapes and surface attributes of these minerals, as they may also affect the likelihood of ice nucleation. This point is partly discussed on page 6, line 5, regarding the study of Whale et al. (2017), but should be explored further. I have included the Kiselev reference below. Additionally, it is mentioned on page 6, line 18, that any relevant mineral textures are difficult to resolve in powdered samples, but these samples can be easily examined in backscattered SEM mode to check for any notable mineral textures. Without knowing the size of the grains, it is difficult to say for sure.

We have added a line referring to the findings of Kiselev et al. (2017) in our discussion of the role of perthite microtexture in ice nucleation by alkali feldspar:

Page 7 Lines 17-21: "[...] the presence of perthitic intergrowth microtexture arising from phase separation (exsolution) into Na- and K-rich regions. Strain at the boundary of these regions gives rise to nanoscale topographic features that are suggested to be important in generating sites for ice nucleation (Whale et al., 2017; Holden et al., 2019). It may be that these features stabilise patches of the high-energy (100) crystallographic plane exposed by surface defects, which Kiselev et al. (2017) showed to be favourable sites for ice nucleation on alkali feldspar."

Unfortunately, in heterogeneous materials such as the studied tephra, it is challenging to isolate individual crystal faces of alkali feldspar to check for such textures. Kiselev et al. (2017), Whale et al. (2017), and Holden

et al. (2019) worked with macroscopic alkali feldspar substrates prepared/oriented to expose particular crystallographic planes (e.g., in the form of thin sections) for examination by high-resolution microscopy. The nature of our powdered samples does not readily allow for this mode of investigation.

**Technical Corrections**

**Line 19 of the abstract: delete the word "partly"**

We have deleted "partly" from this line.

*The sentence beginning on page 2, line 35 just sounds awkward and should be rephrased.*

We have rephrased this sentence:

Page 2 Lines 2-4: "As magma ascends to the surface, the aluminosilicate melt typically carries a cargo of mineral species in the form of crystals suspended within and originating from the melt and/or from the surrounding country rock."

**The third and fourth paragraphs in section 2.1 are not materials or methods and should be placed somewhere else, perhaps the introduction?**

We have moved the content of these paragraphs to the Introduction.

**Page 7, line 21, tephra should be plural.**

We have corrected this.

**Can you discuss further the characteristics of pyroxene minerals that might influence their ice nucleation abilities?**

It is difficult to discuss *"the characteristics of pyroxene minerals that might influence their ice nucleation abilities"* when this information is simply not in the literature. Even the characteristics of (the more comprehensively studied) feldspar minerals that influence their ice nucleation abilities are far from fully understood. We have inserted a few lines to explain that pyroxene is:

Page 7 Lines 40-43/Page 8 Lines 1-2: "[...] an aluminosilicate mineral group of the general formula  $XYZ_2O_6$ , where X and Y are often  $Mg^{2+}$ ,  $Fe^{2+}$  or  $Ca^{2+}$  and Z is  $Si^{4+}$  or sometimes  $Al^{3+}$  (Morimoto et al., 1988). A solid solution exists between the  $Mg_2Si_2O_6$  and  $Fe_2Si_2O_6$  end-members with small amounts of  $Ca^{2+}$  substitution possible (orthopyroxenes), whereas solid immiscibility occurs between other compositions particularly with higher  $Ca^{2+}$  content (clinopyroxenes)."

However, very little research has been done on the ability of pyroxene minerals to nucleate ice, likely because they typically do not occur as a component in desert dust. A very recent study that measured ice nucleation by a clinopyroxene specimen (diopside-augite) concluded that the cause of its INA "is unknown but could be due to any of the variety of mineralogical properties [...] such as crystal lattice match, surface functional group distribution, and topographical features" (Jahn et al., 2019). This warrants investigation and is outside the scope of the current study. We acknowledge (Page 8 Lines 12-13) that "additional research is needed to quantify the INA of a range of pyroxene minerals," to which we have added "and probe the nature of their ice-nucleating properties", and this is in progress in our laboratory (Maters et al., in prep.). We therefore prefer to reserve further discussion of this topic for a future study.

**I don't quite understand the point of plotting both the tephras and glasses in Figure 1a, since they directly overlap in most cases.**

Figure 1b (formerly 1a) illustrates that the nine tephra-glass pairs span a range of compositions in terms of Total Alkali versus Silica classification. Showing that the individual tephra and glass samples in each pair *"directly overlap in most cases"* importantly demonstrates that the two materials are nearly identical in overall chemical composition (i.e., it does not change when the tephra is remelted/quenched to produce the glass).

**When reporting the INE (Tns ~ 1 cm -2) throughout the text, why is the 1 so small...is it subscripted?**

Yes; " $n_s \approx 1 \text{ cm}^{-2}$ " refers to an 'ice-active surface site density approximately equal to one per square centimetre' and, therefore, this entire expression is subscripted to accurately designate that we are referring to the temperature "T" at which this occurs.

**Please state in the caption of Table S1 that these are XRF measurements.**

We have specified in the Table S1 caption that the bulk chemical composition was "determined by X-ray fluorescence".

**Please state in the caption of Table S2 that these are XRF measurements.**

We have specified in the Table S2 caption that the feldspar chemical composition was "determined by electron microprobe analysis" (not by X-ray fluorescence).

**Is the "Text S1" mislabeled? I think the supplementary tables are not currently labeled correctly. Should they be: Table S1 (XRF measurements of bulk samples); Table S2 (Electron microprobe measurements of tephra glasses; Table S3 (Electron microprobe measurements of feldspars).**

The Supplementary Material content is labelled correctly. Table S1 presents X-ray fluorescence measurements of bulk samples (tephras and glasses), Text S1 provides details of the electron microprobe analysis technique, and Table S2 presents electron microprobe measurements of feldspars in the tephras. There are only two tables in the Supplementary Material.

**Figures 4, 6, S1, and S2 need to include error bars or a statement of the errors in the captions.**

We have inserted error bars in Figures 4, 6, S1, and S2. They are very small and typically cannot be seen to extend beyond the data symbols.

**REFERENCES**

Astbury, R. L., Petrelli, M., Arienzo, I., D'Antonio, M., Morgavi, D., and Perugini, D.: Using trace element mapping to identify discrete magma mixing events from the Astroni 6 eruption, in: American Geophysical Union Fall Meeting 2016, San Francisco, United States of America, 12-16 December 2016, 2016AGUFM.V33E3176A, 2016.

Astbury, R. L., Petrelli, M., Ubide, T., Stock, M. J., Arienzo, I., D'Antonio, M., and Perugini, D.: Tracking plumbing system dynamics at the Campi Flegrei caldera, Italy: High-resolution trace element mapping of the Astroni crystal cargo, Lithos, 318-319, 464-477, https://doi.org/10.1016/j.lithos.2018.08.033, 2018.

Atkinson, J. D., Murray, B. J., Woodhouse, M. T., Whale, T. F., Baustian, K. J., Carslaw, K. S., Dobbie, S., O'Sullivan, D., and Malkin, T. L.: The importance of feldspar for ice nucleation by mineral dust in mixed-phase clouds, Nature, 498, 355-358, doi:10.1038/nature12278, 2013.

Augustin-Bauditz, S., Wex, H., Kanter, S., Ebert, M., Niedermeier, D., Stolz, F., Prager, A., and Stratmann, F.: The immersion mode ice nucleation behavior of mineral dusts: A comparison of different pure and surface modified dusts, Geophys. Res. Lett., 41, 7375–7382, doi:10.1002/2014GL061317, 2014.

Ayris, P. M., Lee, A. F., Wilson, K., Kueppers, U., Dingwell, D. B., and Delmelle, P.: SO2 sequestration in large volcanic eruptions: high-temperature scavenging by tephra, Geochim. Cosmochim. Ac., 110, 58-69, https://doi.org/10.1016/j.gca.2013.02.018, 2013.

Ayris, P. M., Delmelle, P., Cimarelli, C., Maters, E. C., Suzuki, Y. J., and Dingwell, D. B.: HCl uptake by volcanic ash in the high temperature eruption plume: Mechanistic insights, Geochim. Cosmochim. Ac., 144, 188-201, https://doi.org/10.1016/j.gca.2014.08.028, 2014.

Connolly, P. J., Möhler, O., Field, P. R., Saathoff, H., Burgess, R., Choularton, T., and Gallagher, M.: Studies of heterogeneous freezing by three different desert dust samples, Atmos. Chem. Phys., 9, 2805–2824, doi:10.5194/acp-9-2805-2009, 2009.D'Oriano, C., Poggianti, E., Bertagnini, A., Cioni, R., Landi, P., Polacci, M., and Rosi, M.: Changes in eruptive style during the A.D. 1538 Monte Nuovo eruption (Phlegrean Fields, Italy): the role of syn-eruptive crystallization, B. Volcanol., 67, 601-621, doi:10.1007/s00445-004-0397-z, 2005.

Delmelle, P., Lambert, M., Dufrêne, Y., Gerin, P., and Óskarsson, N.: Gas/aerosol-ash interaction in volcanic plumes: New insights from surface analysis of fine ash particles, Earth Planet Sc. Lett., 259, 159-170, https://doi.org/10.1016/j.epsl.2007.04.052, 2007.

D'Oriano, C., Poggianti, E., Bertagnini, A., Cioni, R., Landi, P., Polacci, M., and Rosi, M.: Changes in eruptive style during the A.D. 1538 Monte Nuovo eruption (Phlegrean Fields, Italy): the role of syn-eruptive crystallization, B. Volcanol., 67, 601-621, doi:10.1007/s00445-004-0397-z, 2005.

Durant, A. J., Bonadonna, C., and Horwell, C. J.: Atmospheric and environmental impacts of volcanic particulates, Elements, 6, 235-240, doi:10.2113/gselements.6.4.235, 2010.

Durant, A. J., Villarosa, G., Rose, W. I., Delmelle, P., Prata, A. J., and Viramonte, J. G.: Long-range volcanic ash transport and fallout during the 2008 eruption of Chaitén volcano, Chile, Phys. Chem. Earth, 45-46, 50-64, doi:10.1016/j.pce.2011.09.004, 2012.

Genareau, K., Cloer, S. M., Primm, K., Tolbert, M. A., and Woods, T. W.: Compositional and mineralogical effects on ice nucleation activity of volcanic ash, Atmosphere, 9, 238, https://doi.org/10.3390/atmos9070238, 2018.

Harrison, A. D., Lever, K., Sanchez-Marroquin, A., Holden, M. A., Whale, T. F., Tarn, M. D., McQuaid, J. B., and Murray, B. J.: The ice-nucleating ability of quartz immersed in water and its atmospheric importance compared to K-feldspar. In preparation for Atmos. Chem. Phys.

Harrison, A. D., Whale, T. F., Carpenter, M. A., Holden, M. A., Neve, L., O'Sullivan, D., Vergara Temprado, J., and Murray B. J.: Not all feldspars are equal: a survey of ice nucleating properties across the feldspar group of minerals, Atmos. Chem. Phys., 16, 10927-10940, https://doi.org/10.5194/acp-16-10927-2016, 2016.

Heiken, G., and Wohletz, K. (Eds.): Volcanic Ash, University of California Press, London, United Kingdom, 1992.

Herzog, M., Graf, H. F., Textor, C., and Oberhuber, J. M.: The effect of phase changes of water on the development of volcanic plumes. J. Volcanol. Geoth. Res., 87, 55-74, https://doi.org/10.1016/S0377-0273(98)00100-0, 1998.

Holden, M. A., Whale, T. F., Tarn, M. D., O'Sullivan, D., Walshaw, R. D., Murray, B. J., Meldrum, F. C., and Christenson, H. K.: High-speed imaging of ice nucleation in water proves the existence of active sites, Sci. Adv., 5, eaav4316, doi: 10.1126/sciadv.aav4316, 2019.

Hoose, C., and Möhler, O.: Heterogeneous ice nucleation on atmospheric aerosols: a review of results from laboratory experiments, Atmos. Chem. Phys., 12, 9817–9854, doi:10.5194/acp-12-9817-2012, 2012.

Hoshyaripour, G., Hort, M., and Langmann, B.: How does the hot core of a volcanic plume control the sulfur speciation in volcanic emission? Geochem Geophy Geosy, 13, Q07004, https://doi.org/10.1029/2011GC004020, 2012.

Hoshyaripour, G., Hort, M., Langmann, B., and Delmelle, P.: Volcanic controls on ash iron solubility: New insights from high-temperature gas-ash interaction modelling, J. Volcanol. Geoth. Res., 286, 67-77, https://doi.org/10.1016/j.jvolgeores.2014.09.005, 2014.

Jahn, L., Fahy, W., Williams, D. B., and Sullivan, R. C.: The role of feldspar and pyroxene minerals in the ice nucleating ability of three volcanic ashes, ACS Earth Space Chem., XX, XX-XX, doi:10.1021/acsearthspacechem.9b00004, 2019.

Kiselev, A., Bachmann, F., Pedevilla, P., Cox, S. J., Michaelides, A., Gerthsen, D., and Leisner, T.: Active sites in heterogeneous ice nucleation – the example of K-rich feldspars, Science, 355, 367-371, doi:10.1126/science.aai8034, 2017.

Kulkarni, G., Nandasiri, M., Zelenyuk, A., Beranek, J., Madaan, N., Devaraj, A., Shutthanandan, V., Thevuthasan, S., and Varga, T.: Effects of crystallographic properties on the ice nucleation properties of volcanic ash particles, Geophys. Res. Lett., 42, 3048-3055, https://doi.org/10.1002/2015GL063270, 2015.

Kumar, A., Marcolli, C., and Peter, T.: Ice nucleation activity of silicates and aluminosilicates in pure water and aqueous solutions. Part 3 – Aluminosilicates, Atmos. Chem. Phys. Discuss., https://doi.org/10.5194/acp-2018-1021, 2018.

Mangan, T. P., Atkinson, J. D., Neuberg, J. W., O'Sullivan, D., Wilson, T. W., Whale, T. F., Neve, L., Umo, N. S., Malkin, T. L., and Murray, B. J.: Heterogeneous ice nucleation by Soufriere Hills volcanic ash immersed in water droplets, PloS ONE, 12, e0169720, https://doi.org/10.1371/journal.pone.0169720, 2017.

Maters, E. C., Casas, A. S., Dingwell, D. B., Corrado, C., and Murray, B. J.: Alteration of the ice-nucleating effectiveness of volcanic ash by high temperature ash-gas interactions, presented at VMSG 2019 Annual Meeting, St. Andrews, Scotland, 8-10 January 2019.

Maters, E. C., Harrison, A. D., Whale, T. F., and Murray, B. J.: The ice-nucleating activity of various pyroxene minerals in the immersion mode, in preparation for J. Geophys. Res. Atmos.

Morimoto, N., Fabries, J., Ferguson, A. K., Ginzburg, I. V., Ross, M., Seifert, F. A., and Zussman, J.: Nomenclature of pyroxenes, Miner. Petrol., 39, 55-76, https://doi.org/10.1007/BF01226262, 1988.

Parsons, I., Fitz Gerald, J. D., Lee, M. R.: Routine characterization and interpretation of complex alkali feldspar intergrowths, Am. Mineral., 100, 1277-1303, https://doi.org/10.2138/am-2015-5094, 2015.

Peckhaus, A., Kiselev, A., Hiron, T., Ebert, M., and Leisner, T.: A comparative study of K-rich and Na/Ca-rich feldspar ice-nucleating particles in a nanoliter droplet freezing assay, Atmos. Chem. Phys., 16, 11477-11496, doi:10.5194/acp-16-11477-2016, 2016.

Rose, W. I., Delene, D. J., Schneider, D. J., Bluth, G. J. S., Krueger, A. J., Sprod, I., McKee, C., Davies, H. L., and Ernst, G. G. J.: Ice in the 1994 Rabaul eruption cloud: implications for volcano hazard and atmospheric effects, Nature, 375, 477-479, doi:10.1038/375477a0, 1995.

Rose, W. I., Bluth, G. J. S., and Ernst, G. G. J.: Integrating retrievals of volcanic cloud characteristics from satellite remote sensors: a summary, Phil. Trans. R. Soc. Lond. A, 358, 1585-1606, https://doi.org/10.1098/rsta.2000.0605, 2000.

Rose, W. I., Millar, G. A., Mather, T. A., Hunton, D. E., Anderson, B., Oppenheimer, C., Thornton, B. F., Gerlach, T. M., Viggiano, A. A., Kondo, Y., Miller, T. M., and Ballenthin, J. O.: Atmospheric chemistry of a 33–34 hour old volcanic cloud from Hekla Volcano (Iceland): Insights from direct sampling and the application of chemical box modeling, J. Geophys. Res., 111, D20206, doi:10.1029/2005JD006872, 2006.

Schill, G. P., Genareau, K., and Tolbert, M. A.: Deposition and immersion-mode nucleation of ice by three distinct samples of volcanic ash, Atmos. Chem. Phys., 15, 7523-7536, https://doi.org/10.5194/acp-15-7523-2015, 2015.

Van Eaton, A. R., Mastin, L. G., Herzog, M., Schwaiger, H. F., Schneider, D. J., Wallace, K. L., and Clarke, A. B.: Hail formation triggers rapid ash aggregation in volcanic plumes, Nat. Commun., 6, 7860, https://doi.org/10.1038/ncomms8860, 2015.

Welti, A., Lohmann, U., and Kanji, Z. A.: Ice nucleation properties of K-feldspar polymorphs and plagioclase feldspars, Atmos. Chem. Phys. Discuss., https://doi.org/10.5194/acp-2018-1271, 2019.

Whale, T. F., Holden, M. A., Kulak, A. N., Kim, Y.-Y., Meldrum, F. C., Christenson, H. K., and Murray, B. J.: The role of phase separation and related topography in the exceptional ice-nucleating ability of alkali feldspars, Phys. Chem. Chem. Phys., 19, 31186-31193, doi:10.1039/c7cp04898j2, 2017.

Whale, T. F., Holden, M. A., Wilson, T. W., O'Sullivan, D., and Murray, B. J.: The enhancement and suppression of immersion mode heterogeneous ice-nucleation by solutes, Chem. Sci., 9, 4142-4151, doi:10.1039/c7sc05421a, 2018.

Zolles, T., Burkart, J., Häusler, T., Pummer, B., Hitzenberger, R., and Grothe, H.: Identification of ice nucleation active sites on feldspar dust particles, J. Phys. Chem. A, 119, 2692-2700, doi:10.1021/jp509839x, 2015.

---

## Author Response (AR1)

Dr. Elena Maters
School of Earth and Environment
University of Leeds
Woodhouse Lane
Leeds, LS2 9JT
United Kingdom
e.c.maters@leeds.ac.uk

24 March 2019

Dear Dr. Ryan Sullivan,

Thank you for your handling of our manuscript entitled "The importance of crystalline phases in ice nucleation by volcanic ash" (acp-2018-1326) in consideration for publication as a Research Article in *Atmospheric Chemistry and Physics*.

Following up on your request for revisions, we present for your evaluation a detailed reply to the Reviewer comments, as well as a carefully revised manuscript. We have been able to respond fully to all comments and have integrated suggestions to improve the manuscript, or else have explained why they may not be applicable within the context of our study.

The Reviewers express interest in the novelty of our study and confidence in the quality of the data as well as its contribution to the state of knowledge. The main requests related to better placing our work in the context of previous works and acknowledging other interpretations of our results for instance regarding the potential role of pyroxene in nucleating ice. We have now inserted more background on studies of ice nucleation by volcanic ash in the Introduction, and have addressed a very recent study reporting the importance of feldspar and pyroxene in ice nucleation by volcanic ash in the Discussion. We have also revised wording in several places to ensure that plagioclase feldspar and pyroxene are both conveyed as plausible agents responsible for the ice-nucleating activity of the tephra samples that lack alkali feldspar. We recognise that some of our interpretations are speculative and we reiterate that additional studies will be needed to unravel the precise role of mineralogy in ice nucleation by volcanic ash, for example using a combination of pure mineral samples and optical and microanalytical techniques to investigate ice nucleation on a range of solid surfaces relevant to ash.

The other comparatively minor queries raised by the Reviewers have been fully addressed in our reply and/or translated into clarifications in the manuscript or supplement text where necessary.

We thank you for your time and hope that this new submission meets with your approval. We look forward to hearing from you.

Yours sincerely,

Elena Maters

**Authors' response to Reviewers' comments on "The importance of crystalline phases in ice nucleation by volcanic ash" (acp-2018-1326)**

We thank the two Reviewers for their helpful evaluation of our manuscript. We have been able to respond to all comments and provide a carefully revised manuscript. The comments in italic and our responses in normal type are given below. Examples of relevant text (existing or new) are presented in grey highlight between quotation marks. The line numbers that we refer to correspond to those in the revised manuscript.

**REVIEWER #1**

**Major Comments**

*The title of the paper emphasizes the importance of crystalline phases. This, while qualitatively true, is poorly supported by the quantitative techniques in this paper. For example, the LIPteph, LIPglass, CIDteph, and CIDglass are all devoid of crystalline material (<2%, i.e., below the limit-of-detection of the instrument), yet they all have vastly different ice nucleation abilities. In addition, their "glass" examples generally differ from their "teph" case. These are two of the nine cases, so they are over 20% of the samples. Ultimately, that some ice active mineral components (or at least 1400-1600 C labile components) that are present at less than 2% is an interesting result from this work that needs further highlighting.*

We agree with the Reviewer that differences in INA between $LIP_{teph}$, $LIP_{glass}$, $CID_{teph}$, and $CID_{glass}$, which all have <2 wt.% crystalline material, should be addressed given our emphasis of the importance of crystalline phases.

We acknowledge already on Page 4 Lines 8-10 that "While the $LIP_{teph}$ and $CID_{teph}$ crystallinities cannot be quantified below the ~2 wt.% limit of the technique, this does not rule out the possibility of smaller amounts of crystals and/or nanoscale crystallites being present in these samples." Table 2 lists crystalline phases that might occur as minor components below quantification in the tephra, including Fe(-Ti) oxide in $LIP_{teph}$, and alkali feldspar, clinopyroxene, and Fe(-Ti) oxide in $CID_{teph}$. These minor components could be why the two dominantly glassy tephra samples show different INA and are still more ice-active than their counterpart glass samples (Figure 3a,d). We have inserted text in the discussion to highlight this:

Page 5 Lines 33-35: "Even the dominantly glassy $LIP_{teph}$ and $CID_{teph}$ (<2 wt.% crystallinity) display higher INA than their counterpart $LIP_{glass}$ and $CID_{glass}$, which could reflect the influence of minor crystalline components (below quantification by XRD; Table 2) in these tephra on their ability to nucleate ice."

The observation that $CID_{teph}$ is only slightly more ice-active than $CID_{glass}$ (Figure 3d) may, as the Reviewer suggests, reflect some ice-active mineral components also present in amounts <2 wt.% in this glass sample following melting, homogenising, and quenching. This possibility is supported by the observation that $CID_{glass}$ nucleates ice at warmer temperatures than the background water and the other glass samples (Figure 2b,d). We have inserted text in the discussion to highlight this also:

Page 8 Lines 26-28: "It is possible that $CID_{glass}$ and $COL_{glass}$ contain very small amounts of crystals below detection by XRD that survived melting or formed during quenching, which could explain why these glasses stand out in their ability to nucleate ice."

Overall, we think that our findings, both quantitative and qualitative, support the title of the paper.

*Page, 5, Line 26: In this work alone, there is much evidence that Na/Ca-feldspar is not responsible for ice nucleation. For example, the plagioclase parameterizations from Harrison et al. (in prep) are much too low to explain the ice nucleation efficiency of almost all of the ash samples in this work, regardless of their plagioclase content. Furthermore, while Harrison et al., 2016 have shown that higher Na2O/CaO may imply higher ice nucleation activity for some feldspars, Page 5, Line 40 is direct contrast to previous studies that have shown that very pure albite is not that ice nucleation active [e.g., Zolles et al., 2015, Schill et al., 2015,*

*Welti et al., 2019]. Alternatively, while there is less literature evidence that orthopyroxene may be the responsible agent, it seems like a much more feasible choice here. It is interesting to the reviewer that the authors then suggest in the conclusion that intermediate to felsic alkaline magmas may then have a higher propensity to contain ice-active ash in their eruptions.*

We acknowledge that we can improve the presentation of our arguments in this section. The Reviewer is correct that the high INA of tephra samples containing plagioclase feldspar (Page 6 Lines 34-35) "are inconsistent with the relatively low INA of Na-/Ca-feldspar reported in the literature (Fig. 5)." However, evidence of exceptionally ice-active feldspars including a hyper-active Na-feldspar (Amelia albite; characterised by the plagioclase structure) has been presented by Harrison et al. (2016), which we mention and now plot in Figure 5 to illustrate that highly ice-active plagioclase feldspar exists in nature. Even excluding this hyper-active albite, and contrary to the Reviewer's remark that "*albite is not that ice nucleation active*", a new Na-feldspar parameterisation compiled from literature data (Harrison et al., in prep.), which we have also added to Figure 5, falls much closer to the high INA shown by COL$_{teph}$ and TUN$_{teph}$ (Fig. R1 below). We have revised the text as follows:

Page 6 Lines 35-40/Page 7 Lines 1-4: "This may point to the presence in COL$_{teph}$ and TUN$_{teph}$ of ice-active plagioclase feldspar characterised by an INA closer to the Na-feldspar (albite) parameterisation, or potentially more akin to the hyper-active feldspars measured by Harrison et al. (2016; Fig. 5). It is not clear why these hyper-active samples (Amelia albite and TUD#3 microcline) have a much greater INA relative to the majority of feldspars tested (Harrison et al., 2016; Peckhaus et al., 2016), but such wide variability may relate to the specific mechanisms and/or conditions of formation and subsequent processing of individual samples (Welti et al., 2019), and it might be that plagioclase feldspar in COL$_{teph}$ and TUN$_{teph}$ was produced in a way that gives rise to enhanced activity."

Together, the three parameterisations and two hyper-active samples clearly demonstrate the wide variability in INA of feldspar minerals investigated to date. In light of this variability, we maintain our suggestion that highly ice-active plagioclase feldspar might occur in COL$_{teph}$ and TUN$_{teph}$.

[Figure]

**Figure R1.** Ice nucleation active site density ($n_s$) as a function of temperature for 1 wt.% suspensions of tephra in water. The red and green crosses are the $n_s(T)$ values for, respectively, a hyper-active K-feldspar (TUD#3 microcline) and a hyper-active Na-feldspar (Amelia albite) measured by Harrison et al. (2016). The red, green, and blue lines represent parameterisations for, respectively, K-feldspar, Na-feldspar, and Na-/Ca-feldspar reported in Harrison et al. (in prep.) from a compilation of literature data, excluding the hyper-active feldspar specimens. The solid lines indicate mean values and the dashed lines indicate lower and upper limits corresponding to the standard deviation of the mean.

We understand the Reviewer's sentiment "*that orthopyroxene may be the responsible agent*" and we acknowledge (Page 7 Lines 4-5) that the high INA of $COL_{teph}$ and $TUN_{teph}$ alternatively "may relate to the influence of some other mineral component such as orthopyroxene" on ice nucleation by these samples. However, as the Reviewer points out, there is limited literature on ice nucleation by pyroxenes, and so we conclude (Page 8 Lines 11-13) that "we cannot rule out a potential influence of pyroxene on ice nucleation by volcanic ash" and "additional research is needed to quantify the INA of a range of pyroxene minerals, and probe the nature of their ice-nucleating properties, in order to better inform this assessment." We have additionally highlighted a very recent study on ice nucleation by volcanic ash and pyroxene in our discussion:

Page 6 Lines 22-23: "More recently, Jahn et al. (2019) proposed that feldspar and pyroxene were responsible for ice nucleation by ash from Soufrière Hills, Fuego, and Santiaguito volcanoes."

Page 8 Lines 6-11: "More recently, Jahn et al. (2019) measured the INA of a clinopyroxene specimen (freezing from -8 to -24 °C), citing its behaviour to explain the INA of three pyroxene-containing volcanic ash samples. However, XRD analysis indicated that this specimen comprising diopside-augite also contained ~5 wt.% feldspar, which might have influenced the INA observed. In any case, it should be emphasised that a single mineral specimen might not provide a good representation of the INA of a given mineral type, as shown by studies on ice nucleation by feldspar and quartz (Harrison et al., 2016; Whale et al., 2017; Harrison et al., in prep.)."

Our own measurements of a range of pyroxene samples (wollastonite, diopside, augite, enstatite, hypersthene) do not show these minerals to be especially ice-active (Maters et al., in prep.). Therefore, we do not agree that orthopyroxene "*seems like a much more feasible choice*" to explain the high INA of $COL_{teph}$ and $TUN_{teph}$ in our study. However, we recognise that some content in our original manuscript conveyed a preference for the hypothesis of ice-active plagioclase feldspar over that of ice-active orthopyroxene. In light of the Reviewer's comments, and given that we currently lack enough information to conclude definitively which may be the most ice-active mineral(s) in the studied tephra (specifically $COL_{teph}$ and $TUN_{teph}$), we have modified this content so as not to favour either hypothesis:

Page 1 Lines 17-19: "There is evidence of a potential indirect relationship between chemical composition and ash INA, whereby a magma of felsic to intermediate composition may generate ash containing highly ice-active feldspar or pyroxene minerals."

Page 6 Lines 26-28: "[…] while the next most ice-active $COL_{teph}$ and $TUN_{teph}$ are characterised by an abundance of plagioclase feldspar (55 and 43 wt.%, respectively; Fig. 4b) and lesser amounts of orthopyroxene (7 and 5 wt.%, respectively, Fig. 4d)."

Page 8 Lines 33-35: "[…] the $T_{n_s \approx 1\,cm^{-2}}$ decreases with increasing $Fe_2O_3$, MgO, and CaO contents for $NUO_{teph}$, $AST_{teph}$, $COL_{teph}$, $TUN_{teph}$, $ETN_{teph}$, and $KIL_{teph}$ (Fig. 6), in an order consistent with interpretations relating to their feldspar contents and chemistries and/or a potential effect of orthopyroxene in two of these samples (see discussion Sect. 4.2)."

Page 9 Lines 17-20: "[…] we speculate that highly ice-active ash particles might be erupted by volcanoes with intermediate to felsic alkaline magmas giving rise to feldspar crystals featuring overgrowth textures (e.g., Astbury et al., 2016; 2018) or potentially pyroxene crystals with high INA for reasons yet unknown (e.g., Jahn et al., 2019)."

Figure 6: We have inserted the orthopyroxene content of $COL_{teph}$ and $TUN_{teph}$ in the legend.

Lastly, the Reviewer implies that the possibility that (ortho)pyroxene is highly ice-active contradicts our concluding speculation (Page 9 Lines 17-20) that intermediate to felsic alkaline magmas might give rise to highly ice-active ash, presumably because pyroxene can also form in mafic magmas (Figure 1a). However, this

speculation is borne from observations that the most ice-active samples (COL$_{teph}$, TUN$_{teph}$, NUO$_{teph}$, AST$_{teph}$) are those originating from intermediate to felsic magmas (Figure 6), and is not fundamentally affected by whether it is the (ortho)pyroxene, plagioclase feldspar, or alkali feldspar driving ice nucleation by these samples. In contrast, no empirical evidence leads us to speculate that mafic magmas might give rise to highly ice-active ash, as the much less ice-active samples (ETN$_{teph}$, KIL$_{teph}$) containing (clino)pyroxene and/or plagioclase feldspar are those originating from mafic magma (Figure 6). We infer (Page 7 Lines 8-9) that even when tephra samples contain common minerals, "differences [in INA] might relate to the specific chemistry of the mineral phases present in the tephra (Zolles et al., 2015; Welti et al., 2019)," which likely varies depending on factors including magma composition. We thus stand by our original suggestion that intermediate to felsic magmas might erupt highly ice-active crystalline ash.

*Page 6, Line 1. The electron microprobe studies are not described in the text or in the supplemental. I see from Text S1 that the spot size is ~10 um; however, it would be useful to know some quantitative limits on this technique as well as how many spot sizes per sample were looked at.*

The electron microprobe analysis is described in the Supplementary Material, specifically in Text S1, which is referred to by the Reviewer. We have added details regarding the number of spots analysed as well as quantitative limits on this technique:

Supplement Page 3 Lines 2-5: "A 10 µm focused beam was used at an accelerating voltage of 15 keV and a current of 5 nA to analyse at least five points for each crystalline phase in the tephra samples. Elemental detection limits in parts per million are as follows: Si - 786, Al - 655, Fe - 1573, Mg - 501, Ca - 747, Na - 973, K - 711, Ti - 894, Mn - 1401, P - 568, Cr - 1286, S - 767, Cl - 955."

*Page 6, Line 9. I am confused by this paragraph. The authors spend a great deal of time describing why LACteph does not have perthitic intergrowth microtexture, but then end the discussion by stating that NUOteph and ASTteph also don't have perthitic intergrowth microtexture. This seems like a logical fallacy to me. A similar sentiment is felt for the section on the anti-rapakivi texture. Was it not observed in the LACteph sample? Finally, how are all of these these surfaces susceptible to changes upon milling with a zirconia ball and vial?*

We have revised the text in this paragraph to improve our reasoning and have added reference to a very recent study of ice nucleation by various K-feldspars (Welti et al., 2019):

Page 7 Lines 22-39: "However, perthite in alkali feldspar develops in metamorphic and plutonic contexts during slow cooling at temperatures <700 °C (Parsons, 2010), and is generally not expected in volcanic ash which cools rapidly from magmatic down to ambient temperatures during explosive eruption (Parsons et al., 2015). An absence of perthitic microtexture is consistent with the low INA of LAC$_{teph}$ in spite of its alkali feldspar content (9 wt.%). This is supported by evidence that the alkali feldspar mineral sanidine sourced from the same geological setting as LAC$_{teph}$ (Eifel volcanic field) lacks perthitic texture and exhibits a poor ability to nucleate ice (Whale et al., 2017). A recent study similarly found volcanic sanidine from Germany to be the least ice-active among the alkali feldspar samples tested (Welti et al., 2019). In contrast, an absence of perthitic microtexture is inconsistent with the high INA of NUO$_{teph}$ and AST$_{teph}$, and perhaps some other textural feature underlies these samples' ability to nucleate ice as effectively as alkali feldspar of non-pyroclastic origin (Fig. 5). Pyroclastic material from both the 1538 Monte Nuovo eruption (i.e., the origin of NUO$_{teph}$) and the ~4 ka Astroni eruption (i.e., the origin of AST$_{teph}$) has been found to contain anti-rapakivi overgrowth microtexture characterised by plagioclase feldspar cores rimmed by alkali feldspar (D'Oriano et al., 2005; Astbury et al., 2016; 2018). We are not aware of any studies reporting similar textures in pyroclastic material from the 12.9 ka Laacher See eruption (i.e., the origin of LAC$_{teph}$). Such textures are challenging to resolve optically in powdered samples including the tephra studied here. Further optical and microanalytical (Scanning- and Transmission Electron Microscopy) observations (e.g., Whale et al., 2017; Holden et al., 2019) will be needed to explore whether the boundary between Na- and K-rich regions in anti-rapakivi microtexture may give rise to nanoscale topography that induces effects analogous to perthitic microtexture in promoting ice nucleation."

Lastly, the Reviewer queries how such textures might be affected by milling with a zirconia ball and vial. Tephra surfaces are fractured as coarse particles are crushed down to finer particles. However, as we note on Page 7 Lines 17-21, the ability of textures such as perthite to promote ice nucleation is thought to relate to nanoscale topographical features at the boundary of Na- and K-rich phases (Whale et al., 2017; Holden et al., 2019). These nanoscale features would be preserved in milled particles ranging from tens to hundreds of nanometres to several micrometres in diameter.

**Minor/Technical Comments**

*Page 1, Line 10: This abbreviation seems slightly confusing in the context of ice nucleation, since Vali et al. 2014 have proposed the acronym INE as "ice nucleating entity." I suggest you change INE to something like INeff"*

We have changed "INE" for "ice-nucleating effectiveness" to "INA" for "ice-nucleating activity" throughout the manuscript and Supplementary Material.

*Page 1, Line 15: "warmer" instead of higher?*

We have substituted "warmer" into this line.

*Page 1, Line 20: The word "categorically" seems excessive here.*

We have removed "categorically" from this line.

*Page 1, Line 28: The sentence that starts "Ice formation" is unnecessarily long. I would suggest splitting into at least two sentences. A natural break in theme occurs at "as well as."*

We have split this sentence into two.

*Page 1, Line 36: This sentence seems to be missing a comma and coordinating conjunction after diameter. It is the volcanic ash that is usually dominated, not the diameter that is usually dominated.*

We have revised this sentence as follows:

Page 1 Lines 37-39: "By definition, volcanic ash consists of pyroclastic particles <2 mm in diameter, and is comprised of aluminosilicate glass as well as aluminosilicate and/or Fe(-Ti) oxide minerals (Heiken and Wohletz, 1992)."

*Page 1, Line 36: This paragraph seems incredibly weak. Part of the problem is that many statements are weakly lumped into "and references therein." For example, there is little mention of previous work on volcanic ash. For something like dust with hundreds of studies converging on a typical behavior, this could be appropriate–however; there are few previous experiments on volcanic ash, and each of them add holistically to the story presented here.*

We have added content in this paragraph relating to previous studies on volcanic ash ice nucleation:

Page 2 Lines 17-20: "In immersion freezing experiments, Soufrière Hills ash has been found to range from inactive to highly active in nucleating ice, with the discrepancy inferred to relate to differences in ash composition and sample preparation methods (Schill et al., 2015; Mangan et al., 2017; Jahn et al., 2019)."

Page 2 Lines 24-33: "There is increasing evidence that similar factors may influence ice nucleation by volcanic ash. Kulkarni et al. (2015) argued that the presence of amorphous material reduced the INA of Eyjafjallajökull ash compared to Arizona test dust, based on the notion that crystalline structures provide preferred

configurations for water molecules to bind at the particle surface (Pruppacher and Klett, 2010). Schill et al. (2015) proposed that, aside from amorphous versus crystalline content, differences in mineralogy could explain the INA of ash from Soufrière Hills, Fuego, and Taupo volcanoes. Recently, Jahn et al. (2019) suggested that feldspar and pyroxene minerals were responsible for the high INA of Soufrière Hills, Fuego, and Santiaguito ash samples. Genareau et al. (2018) conversely noted a broad trend between chemical composition and ice nucleation, with the INA of five ash samples increasing with $K_2O$ content and decreasing with MnO content."

*Page 2, Line 14: I would prefer that this sentence, if left here, explains how you "improv[ed] the understanding" instead of just simply stating that it will be done.*

We have revised this sentence as follows:

Page 2 Lines 41-42/Page 3 Lines 1-2: "By finding that crystalline phases promote ash ice nucleation and that magma composition may exert an indirect effect via its influence on ash mineralogy, we contribute to an improved understanding of the potential for airborne ash from different eruptions to impact ice formation above the volcanic vent and/or once dispersed in the ambient atmosphere."

*Page 2, Line 22: Since this is an atmospheric chemistry and physics journal, and not a geology journal, it would be helpful for readers of this journal to have the melting point range of each of the minerals in Table 2 compiled for them. That would greatly help them interpret the results of this study and perhaps elucidate why some samples retain some ice nucleation activity after the melt/quench cycle.*

The melting points of minerals in a pure state are different from the melting points of minerals in a heterogeneous mixture such as volcanic tephra. Therefore, it would not be meaningful to report "*the melting point range of each of the minerals in Table 2*", since they would not be reflective of the actual temperature at which the tephra components melted (eutectic melting) during the process of generating the glass samples.

However, the Reviewer makes a good point in implying that the observation that "*some samples retain some ice nucleation activity after the melt/quench cycle*" might be indicative of small amounts of crystals being present in a couple of the glass samples following melting and quenching. We have inserted text to highlight this possibility in explaining the higher INA of CID$_{glass}$ and COL$_{glass}$ relative to the other glasses studied:

Page 8 Lines 26-28: "It is possible that CID$_{glass}$ and COL$_{glass}$ contain small amounts of crystals below detection by XRD that survived melting or formed during quenching, which could explain why these glasses stand out in their ability to nucleate ice."

*Page 3, Line 28 and Line 37: These equations need commas after them, since you have another clause after them (starting with "where").*

We have inserted commas after these equations.

*Page 4, Line 10: I see why these arguments are included in this section, but they seem out of place. For example, the glassy organic particles all nucleated ice in the "deposition" mode and were certainly not immersed in water droplets at or above water saturation. I would suggest the first two sentences of this paragraph be qualified or removed.*

We have removed reference here to ice nucleation by glassy organic particles.

*Page 4, Line 16: As the figure looks now, there does not seem to be significant overlap with the markers and the grayed out region. Perhaps adding error bars to all curves would make that more clear? Or put error bars on the points in Figure 3?*

There is considerable overlap in terms of the temperature range in which freezing of the background water and the glass sample suspensions occurs (not necessarily overlap of individual data points). We have revised the text to clarify this and, as the Reviewer suggests, we have added error bars to the $n_s(T)$ curves of the glass samples in the region of the background water in Figure 2d:

Page 5 Lines 12-15: "For the glass samples in contrast, there is significant overlap of temperatures at which their $f_{ice}(T)$ curves and those of the background water fall, spanning a range from -18 °C to -35 °C (Fig. 2b). As illustrated by the overlap of error bars in their $n_s(T)$ curves (Fig. 2d), this prevents attribution of the observed freezing to ice nucleation by glass particles and comparison of individual glass activities."

*Page 6, Line 28: Or, potentially, mafic magmas with high orthopyroxene?*

Please see our detailed response above to the Reviewer's second major comment, which relates to the same idea. We agree and acknowledge that pyroxene minerals, as well as feldspar minerals, could be responsible for ice nucleation by volcanic ash. However, based on experimental observations regarding the most ice-active tephra samples (COL$_{teph}$, TUN$_{teph}$, NUO$_{teph}$, AST$_{teph}$), we maintain our original speculation in the conclusion that intermediate to felsic alkaline magmas might give rise to crystalline ash that is highly ice-active (Figure 6).

*Page 7, Line 20: There should be a comma between orthopyroxene and which.*

We have inserted a comma here.

*Page 7, Line 37: But–isn't the ash you collected from volcanic plumes where high concentrations of acidic gases likely already interacted with the ash?*

The tephra samples correspond to either ash (COL$_{teph}$, TUN$_{teph}$, AST$_{teph}$) or pumice (LIP$_{teph}$, CID$_{teph}$, NUO$_{teph}$, LAC$_{teph}$, ETN$_{teph}$, KIL$_{teph}$), which to varying extents, likely already interacted with acidic gases and condensates while airborne and may have experienced leaching by water once deposited. However, as we note on Page 3 Lines 11-12: "All samples were crushed to fine powders in a ball mill using a zirconia ceramic ball and vial to ensure consistent treatment of the tephra and glass materials prior to ice nucleation experiments." We have inserted additional text to clarify this in the Materials and Methods:

Page 3 Lines 12-18: "This also reduced the influence of chemically-altered surfaces resulting from tephra interaction with gases and/or liquids post-eruption (e.g., Delmelle et al., 2007), by exposing fresh surfaces with chemical and mineralogical properties reflective of the source magma and any entrained lithic material. This allowed us to address the study objective of assessing specifically the role of chemical composition, crystallinity, and mineralogy on ice nucleation by volcanic ash. Aside from crushing, the samples were not processed by rinsing with water or otherwise, to avoid further alteration of these materials on short time scales (e.g., exposure to water is known to change the INA of some minerals; Harrison et al., 2016; Kumar et al., 2018)."

This is why we acknowledge on Page 9 Lines 28-41 of the Conclusion that - having studied the importance of these primary properties (chemical composition, crystallinity, mineralogy) in ice nucleation by volcanic ash - the next step is to investigate how ash surface 'aging' in the plume and the atmosphere may influence the ash INA. Such further investigations are in progress in our laboratory (e.g., Maters et al., 2019).

*Figure 1 Legend: Is LEI in Figure 1 KIL everywhere else?*

Yes; we have replaced "LEI" in the Figure 1 legend with "KIL" as elsewhere

*Figure 2, 3, and 5. The symbols in these figures are unreasonably small, especially in print form. This makes it relatively difficult to see the different between some of the samples that have similar marker colors (e.g., green, red, light blue.)*

We have amended this; enlarging the symbols in these figures to make it easier to see different sample colours, yet still small enough to distinguish individual sample curves (avoid extensive overlap of closely plotted data points).

**REVIEWER #2**

**Specific Comments**

*Throughout the manuscript you refer to the vertical eruption "plume", and the laterally dispersed "cloud", as defined on line 27 of the introduction. I understand that these terms are not explicitly set by anyone, but physically, it is more correct to call the vertical part the "column" and the laterally spreading part the "plume." Calling the plume a "cloud" is not technically correct and can cause some confusion amongst atmospheric scientists.*

Unfortunately, these terms are not used systematically in the literature. We have adopted terminology consistent with leading authors in the field, where the vertical component connected to the active volcanic vent is called the eruption "plume" (e.g., Herzog et al., 1998; Delmelle et al., 2007; Ayris et al., 2013; 2014; Hoshyaripour et al., 2012; 2014; Van Eaton et al., 2015). Similarly, we call the laterally dispersed component the eruption "cloud", consistent with literature referring to this more dilute feature downwind of the active volcanic vent (e.g., Rose et al., 1995; 2000; 2006; Durant et al., 2010; 2012; Van Eaton et al., 2015).

*Why not refer to the ice nucleation activity (INA) instead of the ice-nucleating effectiveness (INE). INE is already used for other descriptors.*

We have changed "INE" for "ice-nucleating effectiveness" to "INA" for "ice-nucleating activity" throughout the manuscript and Supplementary Material.

*In the Materials and Methods section, more information on the preparation of the tephra samples is required. Were accidental lithics removed? Were the samples rinsed to remove adsorbed salts? This second question relates to the statement on page 7, line 37. Were they altered in any way following eruption? Weathering of the glass postdeposition may introduce small amounts of clay minerals into the samples that are not in high enough quantity to be detected by XRD.*

In short, nothing was done to the tephra samples aside from crushing them in a ball mill. We have added text to explain the reasoning for this in the Materials and Methods:

Page 3 Lines 11-18: "All samples were crushed to fine powders in a ball mill using a zirconia ceramic ball and vial to ensure consistent treatment of the tephra and glass materials prior to ice nucleation experiments. This also reduced the influence of chemically-altered surfaces resulting from tephra interaction with gases and/or liquids post-eruption (e.g., Delmelle et al., 2007), by exposing fresh surfaces with chemical and mineralogical properties reflective of the source magma and any entrained lithic material. This allowed us to address the study objective of assessing specifically the role of chemical composition, crystallinity, and mineralogy in ice nucleation by volcanic ash. Aside from crushing, the samples were not processed by rinsing with water or otherwise, to avoid further alteration of these materials on short time scales (e.g., exposure to water is known to change the INA of some minerals; Harrison et al., 2016; Kumar et al., 2018)."

We have also added text to acknowledge the possibility of soluble salts in the tephra affecting ice nucleation in the Conclusion:

Page 9 Lines 34-39: "Moreover, it has recently been shown that even very low concentrations of soluble salts (1 x 10$^{-4}$ M) can influence the INA of feldspar minerals (Whale et al., 2018; Kumar et al., 2018), and we cannot exclude the possibility that small amounts of NaCl or KCl formed by prior ash-gas/condensate interactions in our tephra samples reduced their INA. However, given the strong correlations observed between INA and composition of the crystalline tephra samples (Fig. 6), we do not think that a potential influence of soluble salts on freezing temperatures affects the general conclusions of this study."

Lastly, weathering of some of the tephra samples post-deposition could have introduced small amounts of clay minerals below detection by XRD, as the Reviewer suggests. However, clay minerals are not thought to be particularly ice-active (e.g., Atkinson et al., 2013; Augustin-Bauditz et al., 2014), and if present in such small quantities are unlikely to have driven the overall trends observed for the crystalline tephra samples, whose INAs are found to correlate well with their bulk compositional and mineralogical properties (Table 2; Figure 6).

*Following milling of the tephras, was a grain size distribution analysis performed to insure the sizes of particles were consistent between samples? Although surface area was measured, the size distribution of the particles may also affect the ice nucleation (i.e., more smaller particles will increase SA compared to fewer, larger particles). Although the milling procedure should effectively homogenize the size distribution, checking this would strengthen the reliability of the results.*

The Reviewer raises the idea that grain size distribution differences between samples could affect ice nucleation since more smaller particles provide greater surface area than fewer larger particles. We agree with this and hence, have normalised the ice nucleation data to the total surface area in each experiment (see Equation 2, Page 4 Lines 31-34), calculated from the sample specific surface area (in m$^2$ g$^{-1}$) determined by nitrogen gas adsorption and the sample mass present in the water droplets (given a 1 wt.% suspension). Expressing the INA of samples in terms of the number of ice nucleation active sites per unit surface area ('$n_s$'), as done widely in literature studies of heterogeneous ice nucleation (e.g., Connolly et al., 2009; Hoose and Möhler, 2012; Zolles et al., 2015; Harrison et al., 2016; Jahn et al., 2019), implicitly accounts for differences in solid surface area provided by particles of different sizes between samples. Therefore, we do not think that there is an added value in presenting the grain size distribution of the milled samples in this study.

*Although they used deposition-mode experiments, and not immersion-mode, the recent study of Kiselev et al. (2017) examined variations in ice nucleation due to defects in the crystal structure of K-feldspars. It may be worthwhile to look at this study for additional information on your interpretations. Specifically, it may be worth considering not only the presence of particular minerals, but the crystal shapes and surface attributes of these minerals, as they may also affect the likelihood of ice nucleation. This point is partly discussed on page 6, line 5, regarding the study of Whale et al. (2017), but should be explored further. I have included the Kiselev reference below. Additionally, it is mentioned on page 6, line 18, that any relevant mineral textures are difficult to resolve in powdered samples, but these samples can be easily examined in backscattered SEM mode to check for any notable mineral textures. Without knowing the size of the grains, it is difficult to say for sure.*

We have added a line referring to the findings of Kiselev et al. (2017) in our discussion of the role of perthite microtexture in ice nucleation by alkali feldspar:

Page 7 Lines 17-21: "[…] the presence of perthitic intergrowth microtexture arising from phase separation (exsolution) into Na- and K-rich regions. Strain at the boundary of these regions gives rise to nanoscale topographic features that are suggested to be important in generating sites for ice nucleation (Whale et al., 2017; Holden et al., 2019). It may be that these features stabilise patches of the high-energy (100) crystallographic plane exposed by surface defects, which Kiselev et al. (2017) showed to be favourable sites for ice nucleation on alkali feldspar."

Unfortunately, in heterogeneous materials such as the studied tephra, it is challenging to isolate individual crystal faces of alkali feldspar to check for such textures. Kiselev et al. (2017), Whale et al. (2017), and Holden

et al. (2019) worked with macroscopic alkali feldspar substrates prepared/oriented to expose particular crystallographic planes (e.g., in the form of thin sections) for examination by high-resolution microscopy. The nature of our powdered samples does not readily allow for this mode of investigation.

**Technical Corrections**

*Line 19 of the abstract: delete the word "partly"*

We have deleted "partly" from this line.

*The sentence beginning on page 2, line 35 just sounds awkward and should be rephrased.*

We have rephrased this sentence:

Page 2 Lines 2-4: "As magma ascends to the surface, the aluminosilicate melt typically carries a cargo of mineral species in the form of crystals suspended within and originating from the melt and/or from the surrounding country rock."

*The third and fourth paragraphs in section 2.1 are not materials or methods and should be placed somewhere else, perhaps the introduction?*

We have moved the content of these paragraphs to the Introduction.

*Page 7, line 21, tephra should be plural.*

We have corrected this.

*Can you discuss further the characteristics of pyroxene minerals that might influence their ice nucleation abilities?*

It is difficult to discuss "*the characteristics of pyroxene minerals that might influence their ice nucleation abilities*" when this information is simply not in the literature. Even the characteristics of (the more comprehensively studied) feldspar minerals that influence their ice nucleation abilities are far from fully understood. We have inserted a few lines to explain that pyroxene is:

Page 7 Lines 40-43/Page 8 Lines 1-2: "[…] an aluminosilicate mineral group of the general formula $XYZ_2O_6$, where X and Y are often $Mg^{2+}$, $Fe^{2+}$ or $Ca^{2+}$ and Z is $Si^{4+}$ or sometimes $Al^{3+}$ (Morimoto et al., 1988). A solid solution exists between the $Mg_2Si_2O_6$ and $Fe_2Si_2O_6$ end-members with small amounts of $Ca^{2+}$ substitution possible (orthopyroxenes), whereas solid immiscibility occurs between other compositions particularly with higher $Ca^{2+}$ content (clinopyroxenes)."

However, very little research has been done on the ability of pyroxene minerals to nucleate ice, likely because they typically do not occur as a component in desert dust. A very recent study that measured ice nucleation by a clinopyroxene specimen (diopside-augite) concluded that the cause of its INA "is unknown but could be due to any of the variety of mineralogical properties […] such as crystal lattice match, surface functional group distribution, and topographical features" (Jahn et al., 2019). This warrants investigation and is outside the scope of the current study. We acknowledge (Page 8 Lines 12-13) that "additional research is needed to quantify the INA of a range of pyroxene minerals," to which we have added "and probe the nature of their ice-nucleating properties", and this is in progress in our laboratory (Maters et al., in prep.). We therefore prefer to reserve further discussion of this topic for a future study.

*I don't quite understand the point of plotting both the tephras and glasses in Figure 1a, since they directly overlap in most cases.*

Figure 1b (formerly 1a) illustrates that the nine tephra-glass pairs span a range of compositions in terms of Total Alkali versus Silica classification. Showing that the individual tephra and glass samples in each pair "*directly overlap in most cases*" importantly demonstrates that the two materials are nearly identical in overall chemical composition (i.e., it does not change when the tephra is remelted/quenched to produce the glass).

*When reporting the INE (Tns ~ 1 cm -2) throughout the text, why is the 1 so small...is it subscripted?*

Yes; "$n_s \approx 1$ cm$^{-2}$" refers to an 'ice-active surface site density approximately equal to one per square centimetre' and, therefore, this entire expression is subscripted to accurately designate that we are referring to the temperature "$T$" at which this occurs.

*Please state in the caption of Table S1 that these are XRF measurements.*

We have specified in the Table S1 caption that the bulk chemical composition was "determined by X-ray fluorescence".

*Please state in the caption of Table S2 that these are XRF measurements.*

We have specified in the Table S2 caption that the feldspar chemical composition was "determined by electron microprobe analysis" (not by X-ray fluorescence).

*Is the "Text S1" mislabeled? I think the supplementary tables are not currently labeled correctly. Should they be: Table S1 (XRF measurements of bulk samples); Table S2 (Electron microprobe measurements of tephra glasses; Table S3 (Electron microprobe measurements of feldspars).*

The Supplementary Material content is labelled correctly. Table S1 presents X-ray fluorescence measurements of bulk samples (tephras and glasses), Text S1 provides details of the electron microprobe analysis technique, and Table S2 presents electron microprobe measurements of feldspars in the tephras. There are only two tables in the Supplementary Material.

*Figures 4, 6, S1, and S2 need to include error bars or a statement of the errors in the captions.*

We have inserted error bars in Figures 4, 6, S1, and S2. They are very small and typically cannot be seen to extend beyond the data symbols.

[revised manuscript text omitted]

felsic ⟵⟶ mafic

▲ NUO tephra (60 wt.% alk. feldspar)
▲ AST tephra (19 wt.% alk. feldspar)
▲ COL tephra (55 wt.% plag. feldspar Na/Ca ≈ 0.5, 7 wt.% orthopyroxene)
▲ TUN tephra (43 wt.% plag. feldspar Na/Ca ≈ 0.5, 5 wt.% orthopyroxene)
▲ ETN tephra (44 wt.% plag. feldspar Na/Ca ≈ 0.2)
▲ KIL teph (3 wt.% plag. feldspar Na/Ca ≈ 0.2)

**Figure 6.** The INA ($T_{n_s \approx 1\ cm^{-2}}$) of NUO$_{teph}$, AST$_{teph}$, COL$_{teph}$, TUN$_{teph}$, ETN$_{teph}$, and KIL$_{teph}$ versus their (a) Fe$_2$O$_3$, (b) MgO, and (c) CaO contents. The grey triangles correspond to tephra samples with comparatively low crystallinities (LIP$_{teph}$, CID$_{teph}$, LAC$_{teph}$) which are excluded from the trendline. Ice nucleation experiments were conducted with 1 wt.% suspensions of tephra in water. The uncertainty in $T_{n_s \approx 1\ cm^{-2}}$ is shown as error bars (note that these are as small as the data symbols). Sample codes are listed in Table 1.

**Supplementary Material**

**Table S1.** Bulk chemical composition of the tephra and glass samples used in this study, determined by X-ray fluorescence and normalised to 100 wt.% (excluding loss on ignition).

| Sample[a] | $SiO_2$ | $Al_2O_3$ | $Fe_2O_3$ | MgO | CaO | $Na_2O$ | $K_2O$ | $TiO_2$ | MnO | $P_2O_5$ |
|---|---|---|---|---|---|---|---|---|---|---|
| *Tephra* | | | | | | | | | | |
| $LIP_{teph}$ | 75.5 | 13.0 | 1.6 | 0.0 | 0.8 | 3.7 | 5.2 | 0.1 | 0.1 | 0.0 |
| $COL_{teph}$ | 61.7 | 18.9 | 4.4 | 1.9 | 5.9 | 4.9 | 1.4 | 0.5 | 0.1 | 0.2 |
| $TUN_{teph}$ | 59.4 | 17.5 | 6.3 | 3.2 | 6.5 | 4.1 | 1.9 | 0.9 | 0.1 | 0.2 |
| $CID_{teph}$ | 62.4 | 17.4 | 4.2 | 0.9 | 1.5 | 7.0 | 5.3 | 0.9 | 0.2 | 0.2 |
| $AST_{teph}$ | 59.5 | 18.9 | 4.2 | 0.9 | 3.2 | 4.0 | 8.6 | 0.5 | 0.1 | 0.2 |
| $NUO_{teph}$ | 60.3 | 19.9 | 3.3 | 0.2 | 1.9 | 6.4 | 7.2 | 0.4 | 0.2 | 0.0 |
| $LAC_{teph}$ | 59.0 | 21.3 | 2.5 | 0.3 | 1.1 | 9.4 | 5.6 | 0.3 | 0.3 | 0.1 |
| $ETN_{teph}$ | 47.7 | 17.3 | 11.3 | 5.2 | 10.4 | 3.6 | 2.0 | 1.7 | 0.2 | 0.6 |
| $KIL_{teph}$ | 50.4 | 13.2 | 12.4 | 8.0 | 10.4 | 2.2 | 0.5 | 2.4 | 0.2 | 0.2 |
| *Glass* | | | | | | | | | | |
| $LIP_{glass}$ | 75.4 | 13.0 | 1.7 | 0.1 | 0.8 | 3.7 | 5.2 | 0.1 | 0.1 | 0.0 |
| $COL_{glass}$ | 61.8 | 18.8 | 4.3 | 2.0 | 5.9 | 4.8 | 1.4 | 0.5 | 0.1 | 0.2 |
| $TUN_{glass}$ | 59.4 | 17.5 | 6.3 | 3.2 | 6.5 | 4.1 | 1.9 | 0.9 | 0.1 | 0.2 |
| $CID_{glass}$ | 62.3 | 17.4 | 4.4 | 0.9 | 1.7 | 6.9 | 5.2 | 0.9 | 0.2 | 0.2 |
| $AST_{glass}$ | 59.6 | 18.9 | 4.2 | 0.9 | 3.2 | 3.9 | 8.5 | 0.5 | 0.1 | 0.2 |
| $NUO_{glass}$ | 60.6 | 20.0 | 3.3 | 0.2 | 1.9 | 6.2 | 7.0 | 0.4 | 0.2 | 0.0 |
| $LAC_{glass}$ | 59.0 | 21.4 | 2.5 | 0.3 | 1.1 | 9.4 | 5.5 | 0.3 | 0.4 | 0.1 |
| $ETN_{glass}$ | 47.7 | 17.4 | 11.2 | 5.2 | 10.4 | 3.6 | 2.0 | 1.7 | 0.2 | 0.6 |
| $KIL_{glass}$ | 50.6 | 13.1 | 12.1 | 7.9 | 10.7 | 2.2 | 0.4 | 2.4 | 0.2 | 0.2 |

[a]Sample codes are listed in Table 1.

[Figure]

**Figure S1.** (a) The difference in INEA ($\Delta T_{n_s \approx 1\ \text{cm}^{-2}}$) between the tephra and glass in each pair versus the crystallinity of the tephra, and (b) the INEA ($T_{n_s \approx 1\ \text{cm}^{-2}}$) of the tephra versus the crystallinity of the tephra. Note that crystallinity below the XRD detectionquantification limit ($\text{LIP}_\text{teph}$, $\text{CID}_\text{teph}$) is plotted at 1 wt.%. Ice nucleation experiments were conducted with 1 wt.% suspensions of tephra or glass in water. The uncertainty in $T_{n_s \approx 1\ \text{cm}^{-2}}$ is shown as error bars (note that these are obscured by the data symbols).

**Text S1.** Electron microprobe analysis of the tephra samples was performed using a Cameca SX-100 instrument equipped with a LaB$_6$ cathode. A 10 µm focused beam was used at an accelerating voltage of 15 keV and a current of 5 nA to analyse at least five points for each crystalline phase in the tephra samples. Calibration was done on the following standard materials: albite - Na, Si; periclase - Mg; orthoclase - K, Al; wollastonite - Ca, Si; Fe$_2$O$_3$ - Fe; Cr$_2$O$_3$ - Cr; ilmenite - Ti; bustamite - Mn; apatite - P; vanadinite - Cl; anhydrite - S. Elemental detection limits in parts per million are as follows: Si - 786, Al - 655, Fe - 1573, Mg - 501, Ca - 747, Na - 973, K - 711, Ti - 894, Mn - 1401, P - 568, Cr - 1286, S - 767, Cl - 955.

**Table S2.** Average chemical composition of feldspar in tephra samples used in this study, determined by electron microprobe analysis and expressed in wt.%.

| Sample[a] | | SiO$_2$ | Al$_2$O$_3$ | Fe$_2$O$_3$ | MgO | CaO | Na$_2$O | K$_2$O | TiO$_2$ | MnO | P$_2$O$_5$ | Cr$_2$O$_3$ | SO$_3$ | Cl | Total | Na$_2$O/CaO in *pl* | K$_2$O/Na$_2$O in *al* |
|---|---|---|---|---|---|---|---|---|---|---|---|---|---|---|---|---|---|
| COL$_{teph}$ | *pl* | 54.2 | 28.1 | 0.76 | 0.05 | 10.8 | 5.6 | 0.22 | <d.l. | <d.l. | <d.l. | - | - | - | 99.7 | 0.5 | - |
| TUN$_{teph}$ | *pl* | 55.7 | 27.4 | 1.1 | 0.10 | 10.9 | 5.3 | 0.44 | 0.09 | <d.l. | <d.l. | <d.l. | <d.l. | <d.l. | 101.2 | 0.5 | - |
| AST$_{teph}$ | *pl* | 54.6 | 26.9 | 0.68 | <d.l. | 9.6 | 4.7 | 2.5 | 0.04 | <d.l. | <d.l. | <d.l. | - | <d.l. | 99.1 | 0.5 | - |
| | *al* | 63.7 | 19.3 | 0.40 | <d.l. | 0.84 | 2.5 | 12.4 | 0.10 | <d.l. | - | <d.l. | <d.l. | - | 99.4 | - | 5.0 |
| NUO$_{teph}$ | *al* | 64.0 | 20.7 | 0.85 | <d.l. | 2.1 | 5.9 | 7.0 | 0.16 | <d.l. | <d.l. | <d.l. | <d.l. | <d.l. | 100  | - | 1.2 |
| LAC$_{teph}$ | *al* | 63.9 | 20.2 | 0.76 | 0.05 | 1.3 | 4.8 | 9.2 | 0.14 | <d.l. | <d.l. | <d.l. | <d.l. | <d.l. | 100.4 | - | 1.9 |
| ETN$_{teph}$ | *pl* | 48.5 | 32.4 | 1.2 | 0.08 | 15.9 | 2.5 | 0.22 | 0.09 | <d.l. | <d.l. | <d.l. | <d.l. | - | 100.9 | 0.2 | - |
| KIL$_{teph}$ | *pl* | 50.3 | 31.0 | 1.0 | 0.17 | 15.4 | 3.0 | 0.12 | 0.11 | - | <d.l. | - | <d.l. | - | 101.2 | 0.2 | - |

[a]Sample codes are listed in Table 1. *pl* = plagioclase (Na-/Ca-) feldspar, *al* = alkali (K-) feldspar. d.l. = detection limit.

[Figure]

**Figure S2.** The INEA ($T_{n_s \approx 1\ cm^{-2}}$) of the tephra (red triangles) and glass (blue circles) versus their (a) SiO$_2$, (b) Al$_2$O$_3$, (c) Fe$_2$O$_3$, (d) MgO, (e) CaO, (f) Na$_2$O, (g) K$_2$O, (h) TiO$_2$, (i) MnO, and (j) P$_2$O$_5$ contents. The open blue circles correspond to glasses (all except CID$_{glass}$ and COL$_{glass}$) for which ice nucleation cannot be distinguished from that induced by the background water. Ice nucleation experiments were conducted with 1 wt.% suspensions of tephra or glass in water. The uncertainty in $T_{n_s \approx 1\ cm^{-2}}$ is shown as error bars (note that these are obscured by the data symbols).